# LVLM-eHub: A Comprehensive Evaluation Benchmark for Large Vision-Language Models

## Abstract

Large Vision-Language Models (LVLM) have recently played a dominant role in multimodal vision-language learning. Despite the great success, it lacks a holistic evaluation of their efficacy. This paper presents a comprehensive evaluation of publicly available large multimodal models by building an LVLM evaluation Hub (LVLM-eHub). Our LVLM-eHub consists of 8 representative LVLMs such as InstructBLIP and MiniGPT-4, which are thoroughly evaluated by a quantitative capability evaluation and an online arena platform. The former evaluates 6 categories of multimodal capabilities of LVLMs such as visual question answering and embodied artificial intelligence on 40 standard text-related visual benchmarks, while the latter provides the user-level evaluation of LVLMs in an open-world question-answering scenario. The study reveals several innovative findings. First, Instruction-tuned LVLM with massive in-domain data such as InstructBLIP may overfit many existing tasks, generalizing poorly in the open-world scenario. Second, Instruction-tuned LVLM with moderate instruction-following data may result in object hallucination issues (i.e., generate objects that are inconsistent with target images in the descriptions). It either makes the current evaluation metric such as CIDER for image captioning ineffective or generates wrong answers. Third, employing a multi-turn reasoning evaluation framework could mitigate the issue of object hallucination, shedding light on developing an effective metric for LVLM evaluation. The findings provide a foundational framework for the conception and assessment of innovative strategies aimed at enhancing zero-shot multimodal techniques. The evaluation pipeline will be available at vlarena page.

## 1 Introduction

Large Language Models (LLMs), such as LLaMA [1], GPT-3 [2], and Vicuna [3], have demonstrated remarkable progress in Natural Language Processing (NLP). These models leverage large-scale pre-training data and huge networks to achieve impressive results in NLP benchmarks. Recently, GPT-4 [4] further expanded the impact to the multimodal community, stimulating the rapid development of large vision-language models (LVLMs) and revolutionizing the landscape of artificial intelligence.

Large Vision-Language Models (LVLM) have achieved remarkable progress in multimodal vision-language learning for various multimodal tasks such as visual question answering and multimodal conversation. Specifically, LVLMs capitalize on the knowledge from LLMs and effectively align visual features with the textual space. Flamingo [5], a pioneering LVLM, integrates visual features into LLMs through cross-attention layers. Later studies proposed more efficient vision-text interactions [6], more efficient training methods [7, 8], and employing instruction tuning [9, 7, 9, 10, 11, 12, 13, 8].

However, despite the great success, few efforts have been made to provide systematic evaluations of LVLMs. But evaluation plays a critical role in understanding the strengths and weaknesses of LVLMs,

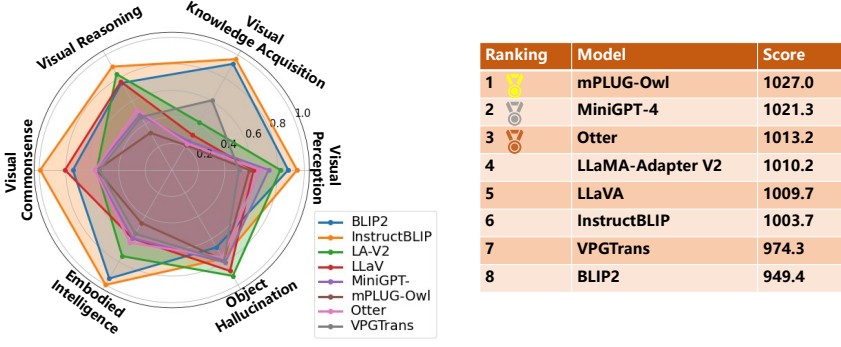

**(a) Quantitative Capability Evaluation**          **(b) LVLMs Arena Ranking**

Figure 1: Comparative analysis of LVLMs within the LVLM eHub. (a) illustrates the variances in quantitative capability performance across six distinct aspects among LVLMs. (b) presents the Elo rating ranking of LVLMs within the LVLM Arena.

thereby guiding their future development. Recent work [14] presents a systematic investigation of object hallucination of LVLMs by proposing a polling-based object probing evaluation method. Moreover, ImageNetVC [15] studies how well LVLMs can master visual commonsense knowledge. Liu et al. [16] comprehensively evaluate the performance of LVLMs in visual recognition with text recognition, such as optical character recognition. GVT [17] evaluates LVLM's visual semantic understanding and fine-grained perception capabilities. Nevertheless, these studies only evaluate a portion of LVLMs on specific tasks, lacking an overall understanding of LVLM's capabilities.

In pursuit of a comprehensive evaluation of LVLMs, we build an LVLM Evaluation hub (LVLM-eHub) consolidating 8 representative LVLMs such as InstrucBLIP and MiniGPT-4. The detailed information about model configuration and training data is listed in Table 1. Our LVLM-eHub consists of a quantitative capability evaluation and an online arena platform, providing a thorough investigation of the selected LVLMs. Specifically, the quantitative capability evaluation extensively evaluates 6 categories of multimodal capabilities of LVLMs including visual perception, visual knowledge acquisition, visual reasoning, visual commonsense, object hallucination, and embodied intelligence (see Fig. 1 (a)), by collecting 40 standard text-related visual benchmarks. On the other hand, the online arena platform features anonymous randomized pairwise battles in a crowd-sourced manner, providing a user-level model ranking in the open-world question-answering scenario (see Fig. 1 (b)).

Our LVLM-eHub comprehensively evaluates LVLMs, revealing several innovative findings. (1) Instruction-tuned LVLM with massive in-domain data suffers from overfitting and generalizes poorly in open-world scenarios, such as InstructBLIP (see Fig. 1 (a)). (2) With moderate instruction-following data, Instruction-tuned LVLM may cause object hallucination issues, generating objects that are inconsistent with target images in the descriptions. This leads to incorrect answers or renders current evaluation metrics, such as CIDER for image captioning, ineffective. (3) We find that a multi-turn reasoning evaluation pipeline can mitigate the issue of object hallucination, indicating that developing an effective metric for LVLM evaluation is urgent.

The contributions of our work are summarized follows. (1) We propose LVLM-eHub which is the first comprehensive evaluation benchmark for large vision-language models, to our best knowledge. (2) LVLM-eHub provides extensive evaluation on 6 categories of multimodal capabilities of LVLMs in more than 40 text-based visual tasks. (3) LVLM-eHub builds an online arena platform for LVLMs, which features anonymous randomized pairwise user-level comparison in a open-world scenario. (4) Our evaluation results reveal several innovative findings, providing a foundational framework for the assessment of innovative strategies aimed at enhancing zero-shot multimodal techniques.

## 2 LVLM Evaluation Hub

In this section, we introduce representative LVLMs, multimodal capabilities of interest, and evaluation methods. The whole LVLM Evaluation Hub is illustrated in Fig. 2. Our LVLM evaluation hub

| Model | Model Configuration | | | | | | Image-Text Data | | Visual Instruction Data | |
|---|---|---|---|---|---|---|---|---|---|---|
| | VE | LLM | Adapt | ToP | TuP | # Token | Source | Size | Source | Size |
| BILP2 | ViT-g/14$^\dagger$ | FlanT5-XL$^\dagger$ | Q-Former | 4B | 107M | 32 | CC$^*$-VG-SBU-L400 | 129M | - | - |
| LLaVA | ViT-L/14$^\dagger$ | Vicuna | FC layer | 7B | 7B | 256 | CC3M | 595K | LLaVA-I | 158K |
| LA-V2 | ViT-L/14$^\dagger$ | LLaMA$^\dagger$ | B-Tuning | 7B | 63.1M | 10 | L400 | 200M | LLaVA-I+G4L | 210K |
| MiniGPT-4 | BLIP2-VE$^\dagger$ | Vicuna$^\dagger$ | FC layer | 7B | 3.1M | 32 | CC-SBU-L400 | 5M | CC+ChatGPT | 3.5K |
| mPLUG-Owl | ViT-L/14 | LLaMA$^\dagger$ | LoRA | 7B | 1.1B | 65 | CC$^*$-CY-L400 | 204M | LLaVA-I | 158K |
| Otter | ViT-L/14$^\dagger$ | LLaMA$^\dagger$ | Resampler | 9B | 1.3B | 64 | - | - | LLaVA-I | 158K |
| InstructBLIP | ViT-g/14$^\dagger$ | Vicuna$^\dagger$ | Q-Former | 7B | 107M | 32 | - | - | QA$^*$ | 16M |
| VPGTrans | ViT-g/14$^\dagger$ | Vicuna$^\dagger$ | Q-Former | 7B | 107M | 32 | COCO-VG-SBU | 13.8M | CC+ChatGPT | 3.5K |

Table 1: **Comparison of Different LVLMs.** 'VE', 'Adapt', 'ToP', 'TuP', and '# Token' represent the visual encoder, adaption module, number of total parameters, tuning parameters, and visual tokens fed into the text encoder, respectively. $^\dagger$ indicates that the model is frozen. CC$^*$ consists of COCO [18], CC3M [19], and CC12M [20]. CC, VG, SBU CY, and L400 indicate Conceptual Caption [19, 20], Visual Genome [21], COYO-700M [22] and LAION 400M [23], respectively. LLaVA-I and G4L represent 158K multimodal instruction-following data in LLaVA [9] and data generated by GPT-4 for building an instruction-following LLMs [24]. QA$^*$ denotes 13 question-answering datasets in InstructBLIP [13]. We count all the data and tuning parameters needed to convert the pretrained vision model and LLM into a visual instruction model. The average score is obtained by normalizing over each row and taking the average of each column.

compromises 8 representative models including BLIP2 [6], LLaVa [9], LLaMA-Adapter V2 [7], MiniGPT-4 [10], mPLUG-Owl [11], Otter [12], InstructBLIP [13], and VPGTrans [8]. All models boost vision-language representation learning by utilizing pre-trained image encoders and large language models (LLM). But they differ in training data scale and model configuration as shown in Table 1. For a fair comparison between LVLMs, we collect their checkpoints with parameter sizes less than 10B. The detailed descriptions of these models are in the Appendix.A.

## 2.1 Quantitative Capability Evaluation

We aim to evaluate LVLMs' capability comprehensively. In particular, we summarize 6 categories of capabilities and collect corresponding benchmarks for quantitative evaluation (see Fig.2). Please see our supplementary materials for more statistics and details of the collected benchmarks.

**Visual Perception.** Visual perception is the ability to recognize the scene or objects in images, the preliminary ability of the human visual system. We evaluate this capability of models through image classification (ImgCLs) using the ImageNet1K [25], CIFAR10 [26], Pets37 [27] and Flowers102 [28] benchmarks, multi-class identification (MCI) and object counting (OC) using the GVT [29] benchmark. ImgCLs and MCI measure how well an LVLM grasps high-level semantic information, while OC assesses the recognition ability for fine-grained objects.

**Visual Knowledge Acquisition.** Visual knowledge acquisition entails understanding images beyond perception to acquire knowledge. This evaluation is conducted through Optical Characters Recognition (OCR) using twelve benchmarks (including IIIT5K [30], IC13 [31], IC15 [32], Total-Text [33], CUTE80 [34], SVT [35], SVTP [36], COCO-Text [37], WordArt [38], CTW [39], HOST [40], WOST [40]), Key Information Extraction (KIE) using the SROIE [41] and FUNSD [42], and Image Captioning (ImgCap) using two benchmarks (including NoCaps [43] and Flickr30K [44]). The OCR task measures whether a model can accurately identify and extract text from images or scanned documents. The KIE task further poses challenges in extracting structured information from unstructured or semi-structured text. Finally, ImgCap assesses whether a model can generate a good natural language description of the content of an image.

**Visual Reasoning.** Visual reasoning requires a comprehensive understanding of images and related texts. To evaluate the visual reasoning ability of LVLMs, we utilize three tasks including visual question answering (VQA), knowledge-grounded image description (KGID), and visual entailment SNLI-VE [45]), two benchmarks (i.e. ScienceQA [46] and VizWiz [47] ) and one benchmark (i.e. SNLI-VE), respectively. These three tasks are in VQA form in different domains. A capable LVLM should be able to understand the objects and scenes in an image and can reason to generate answers that are semantically meaningful and relevant to the question asked.

**Visual Commonsense.** Visual commonsense refers to the general visual knowledge commonly shared across the world, as opposed to the visual information specific to a single image. This evaluation tests

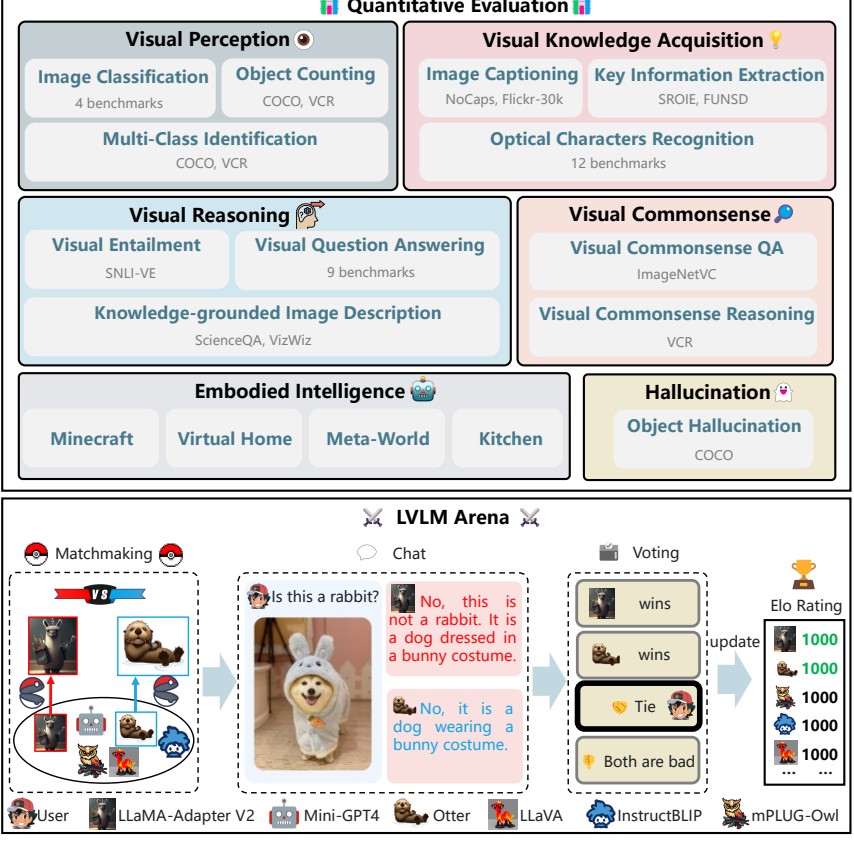

Figure 2: Our evaluation encompasses quantitative evaluation and online LVLM Arena. Plentiful benchmarks are employed to comprehensively evaluate the six critical capabilities of the models in the quantitative evaluation. In the LVLM Arena, an online platform, users can participate in an online evaluation by chatting with two anonymous models and choosing their preferred model.

the model's understanding of commonly shared human knowledge about generic visual concepts using ImageNetVC [15] and visual commonsense reasoning (VCR) [48]. Specifically, ImageNetVC is utilized for zero-shot visual commonsense evaluation, such as color and shape, while VCR covers various scenes, such as spatial, casual, and mental commonsense.

**Embodied Intelligence.** Embodied intelligence aims to create agents, such as robots, which learn to solve challenging tasks requiring environmental interaction. Recently, LLM and LVLM exhibited exceptional effectiveness in guiding the agent to complete a series of tasks. In this evaluation, we utilize high-level tasks as in EmbodiedGPT [49] and employ Minecraft [50], VirtualHome [51], Meta-World [52], and Franks Kitchen [52] as benchmarks.

**Object Hallucination.** It is known that LVLM suffers from the object hallucination problem, i.e., the generated results are inconsistent with the target images in the descriptions [14]. Evaluating the degree of object hallucination for different LVLMs help understand their respective weaknesses. To this end, we evaluate the object hallucination problem of LVLMs on the MSCOCO dataset [18].

## 2.2 Online Evaluation with LVLM Arena

Designing quantitative evaluations for LVLM to satisfy all capabilities is challenging, as evaluating LVLM responses constitutes an open-ended problem. Inspired by FastChat [53], we introduce the LVLM Arena, an online evaluation framework for LVLMs' pairwise battle with human judgment.

Figure 2 illustrates the LVLM Arena, comprising three primary components: matchmaking, chat, and voting. Initially, two models are sampled from the model zoo. Users then converse side-by-side with the models, who remain anonymous. Subsequently, users vote for the superior model.

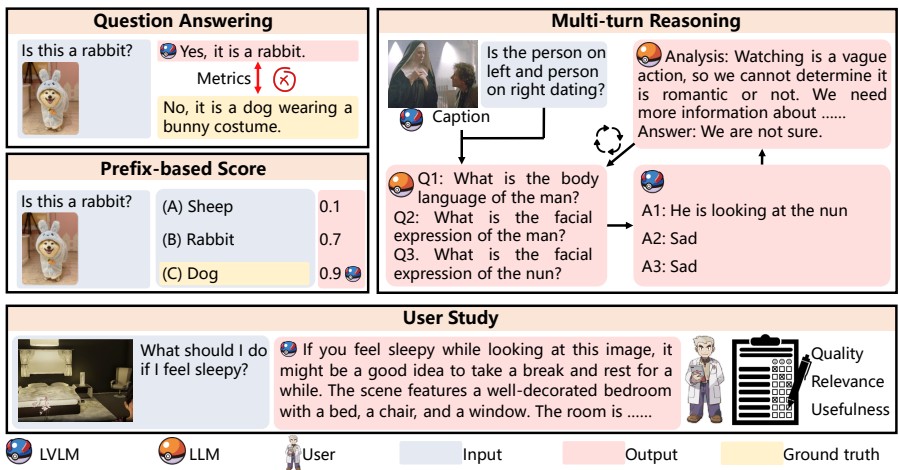

Figure 3: Illustration of our adopted evaluation methods. To evaluate the zero-shot performance of LVLMs on diverse downstream tasks, we employ four methods including question answering, prefix-based score, multi-turn reasoning, and user study.

**Matchmaking.** The matchmaking module samples two models in a tournament style based on their Elo rating. However, due to the currently limited size of the model hub, we employ random sampling.

**Chat.** Users chat side-by-side with two sampled models (which remain anonymous) using images or text inputs. Different from quantitative evaluation, users can chat about anything. Our existing online platform supports only single-round chats due to multi-round chats' high computational and memory demands. Future updates will address this constraint.

**Voting.** After the chat session, users vote for their preferred model. Four options are available: Model A, Model B, Tie, and Both are bad. The Elo rating is subsequently updated using voting results.

In contrast to limited quantitative evaluations, the LVLM Arena provides an open-world evaluation framework that enables users to chat with models about anything, emulating real-world conditions. Besides, users serve as the judge for the battle, which brings more convincing evaluation results than traditional evaluation metrics.

## 2.3 Zero-shot Evaluation

LVLMs are capable of capturing a wide range of multimodal patterns and relationships. We evaluate the above 6 categories of capabilities of LVLMs by investigating their zero-shot performance on various tasks. Zero-shot evaluation allows us to evaluate the LVLMs' ability to generalize to new tasks without training the model, which is competent for large-scale evaluation. To be specific, we treat the zero-shot evaluation as various forms of prompt engineering for different tasks (see Fig. 3) as presented in the following.

- *Question Answering.* Prompting with visual question answering can be used to solve many downstream tasks, which assess how well an LVLM understands the underlying language and visual features. We design proper prompts to ensure that the LLM can produce meaningful results. For example, text prompts of OCR can be "*what is written in the image?*". Then, we evaluate the answers generated by the LLM using the corresponding metric such as accuracy.
- *Prefix-based Score.* For multi-choice QA tasks, we can utilize a visual encoder to obtain visual prompts for a given image. Then, the visual prompts are prefixed into the text embeddings, which are fed into the LLM. The likelihood of image-text pair can be generated, which is referred to as a prefix-based score. We can obtain a prefix-based score for each text prompt of the candidate's answer. The answer with the largest prefix-based score is selected as the final answer. We provide the formulation in Sec. A.3 of Appendix.
- *Multi-turn Reasoning.* Following IdealGPT [16], we use a multi-turn reasoning framework to evaluate complex visual reasoning tasks. Specifically, we utilize an LLM such as ChatGPT to generate sub-questions for a given question, an LVLM to provide corresponding sub-answers, and

| | Datasets | BLIP2 | InstructBLIP | LA-V2 | LLaVA | MiniGPT-4 | mPLUG-Owl | Otter | VPGTrans | S-SOTA |
|---|---|---|---|---|---|---|---|---|---|---|
| ImgCls | ImageNet1K [54] | 23.71 | 24.51 | 25.89 | 23.50 | 21.58 | **26.81** | 19.29 | 15.60 | 91.10 [55] |
| | CIFAR10 [26] | 58.20 | 67.24 | 64.86 | **67.96** | 61.17 | 53.09 | 65.42 | 53.11 | 99.70 [56] |
| | Pets37 [27] | 34.83 | **39.17** | 24.56 | 9.05 | 19.81 | 33.66 | 5.91 | 8.56 | 96.70 [57] |
| | Flowers102 [28] | 30.90 | **32.79** | 32.05 | 11.99 | 29.74 | 20.15 | 10.41 | 10.46 | 99.64 [58] |
| OC | COCO | **48.90** | 46.65 | 38.50 | 20.56 | 20.86 | 27.51 | 46.14 | 25.46 | - |
| | VCR | 25.05 | 29.29 | 26.51 | 24.60 | 25.26 | 8.99 | **41.06** | 18.03 | - |
| MCI | COCO | 86.06 | **87.81** | 82.90 | 49.66 | 72.70 | 35.39 | 51.03 | 50.98 | - |
| | VCR | 66.59 | **76.49** | 50.66 | 66.90 | 66.02 | 19.12 | 51.60 | 47.13 | - |
| Avg. | | 0.879 | **0.946** | 0.820 | 0.617 | 0.731 | 0.753 | 0.669 | 0.507 | - |

Table 2: Evaluation results of visual perception capability of LVLMs on tasks of Image Classification (Imgcls), Object Counting (OC), and Multi-class Identification (MCI). The **best** result is **bold** while the second is underlined. S-SOTA indicates the supervised state-of-the-art results

another LLM to reason to assess sub-answers' quality. Such a pipeline iteratively proceeds until a satisfactory answer is obtained. We provide the formulation in Sec. A.3 of Appendix.

• *User Study.* Evaluating the quality of the text generated by an LVLM requires a thorough understanding of the underlying language and context. In embedded artificial intelligence tasks, the LVLM generates a plan for the given instruction, which should be evaluated through various aspects such as recognition accuracy and conciseness in answers. It is hard to implement such an evaluation using an existing metric. Thus, user studies are conducted to assess the quality, relevance, and usefulness of the text generated by the LVLM in a specific context. To maintain evaluation fairness, we randomly shuffle the model's output order and anonymize outputs during evaluation.

Note that our user study does not involve direct interactions with human participants and does not involve potential risks to participants, such as the collection of personal information, or any other aspects that could impact the participants' rights or well-being. Currently, we do not include an IRB Approval. We are dedicated to addressing the ethical and moral considerations regarding the user evaluation method with thoroughness and commitment, while also providing effective solutions.

# 3 Experiment and Analysis

In this section, we perform a zero-shot evaluation to assess the 6 kinds of capabilities of LVLMs. Specifically, visual perception ability, visual knowledge acquisition, visual Reasoning, visual commonsense understanding, visual object hallucination, and embodied intelligence are assessed in Sec. 3.1 ∼ Sec.3.6, respectively. The LVLM arena evaluation result is presented in Sec.3.7. More quantitative results can be found in Appendix C.

## 3.1 Results on Visual Perception

Visual perception is an important ability of LVLMs. As presented in Sec. 2.1, we evaluate through image classification (ImgCls), multi-class identification (MCI), and object counting (OC). The evaluation details of tasks are demonstrated in Appendix.B.1. The evaluation results are reported in Table 2. We have three observations. (1) mPLUG-Owl and LLaVA perform best on coarse-grained classification tasks (*i.e.,* ImageNet1K and CIFAR10). The commonality is that they update LLM with 158K instruction-following data. (2) InstructBLIP presents good perception ability in fine-grained ImgCls, OC, and MCI tasks. The main reason is that InstructBLIP may be fine-tuned on various existing VQA datasets, which may make it overfit on these tasks. (3) The performances of LVLMs on ImgCls are significantly inferior to supervised SOTA, indicating plenty of room for LVLM's perception ability.

## 3.2 Results on Visual Knowledge Acquisition

Visual knowledge acquisition involves going beyond image perception to acquire deeper understanding and knowledge. In our study, we evaluate the acquisition of visual knowledge through various tasks, namely Optical Character Recognition (OCR), Key Information Extraction (KIE), and Image Captioning, all performed in a Visual Question Answering (VQA) fashion. The evaluation details of tasks are demonstrated in Appendix.B.2. Table 3 shows the zero-shot performance in visual knowledge acquisition, and we have the following observations. First, BLIP2, InstructBLIP, and

| Datasets | | BLIP2 | InstructBLIP | LA-V2 | LLaVA | MiniGPT-4 | mPLUG-Owl | Otter | VPGTrans | S-SOTA |
|---|---|---|---|---|---|---|---|---|---|---|
| OCR | IIIT5K | 80.17 | **83.90** | 36.30 | 31.57 | 25.13 | 26.50 | 17.57 | 51.50 | 99.2[59] |
| | IC13 | 81.13 | **82.08** | 20.87 | 16.39 | 16.75 | 14.86 | 09.67 | 61.67 | 98.4[60] |
| | IC15 | 66.68 | **73.57** | 29.40 | 26.58 | 21.43 | 21.14 | 18.49 | 42.00 | 91.4[59] |
| | Total-Text | 68.31 | **71.51** | 30.93 | 24.51 | 18.65 | 21.08 | 14.81 | 43.60 | 90.5[61] |
| | CUTE80 | 85.07 | **86.11** | 35.76 | 36.46 | 33.33 | 34.03 | 18.75 | 62.85 | 99.3[59] |
| | SVT | 85.78 | **86.86** | 20.40 | 18.55 | 17.47 | 13.45 | 10.51 | 51.16 | 98.3[59] |
| | SVTP | 77.34 | **80.93** | 31.01 | 27.44 | 19.69 | 20.78 | 19.22 | 47.13 | 97.2[59] |
| | COCO-Text | 53.62 | **58.25** | 20.94 | 18.05 | 12.05 | 13.50 | 11.30 | 27.00 | 81.1[59] |
| | WordArt | 73.66 | **75.12** | 38.98 | 35.87 | 31.57 | 32.36 | 21.05 | 53.30 | 72.5[38] |
| | CTW | 67.43 | **68.58** | 18.13 | 16.73 | 15.14 | 12.91 | 10.05 | 40.80 | 88.3[61] |
| | HOST | 57.28 | **61.22** | 16.60 | 15.94 | 14.57 | 11.92 | 10.14 | 32.20 | 77.5[59] |
| | WOST | 68.83 | **73.26** | 21.73 | 20.49 | 17.47 | 14.45 | 12.29 | 37.91 | 87.5[59] |
| KIE | SROIE | 0.08 | **0.09** | 0.02 | 0.01 | 0.01 | 0.01 | 0.01 | 0.06 | 97.81[62] |
| | FUNSD | 1.02 | 1.03 | **2.16** | 1.93 | 1.20 | 0.41 | 1.91 | 1.27 | 89.45[63] |
| Image Captioning | NoCaps | **48.60** | 46.61 | 33.69 | 1.56 | 5.84 | 0.26 | 11.56 | 36.20 | 124.77[64] |
| | Flickr-30k | 46.65 | **50.69** | 23.85 | 2.23 | 2.66 | 0.02 | 7.12 | 23.41 | - |
| Average Score | | 0.924 | **0.965** | 0.416 | 0.307 | 0.253 | 0.215 | 0.231 | 0.607 | - |

Table 3: Comparison of Zero-shot Performance for Large-scale Vision and Language Models (LVLMs) on OCR, KIE, and Image Captioning Tasks. Evaluation metrics include word accuracy for OCR datasets, entity-level F1 score for KIE datasets, and CIDEr score for image captioning datasets.

| Datasets | | BLIP2 | InstructBLIP | LLaMA-Adapter-v2 | LLaVA | MiniGPT-4 | mPLUG-Owl | Otter | VPGTrans | S-SOTA |
|---|---|---|---|---|---|---|---|---|---|---|
| VQA | DocVQA | 4.75 | 5.89 | **8.13** | 6.26 | 3.57 | 2.24 | 3.44 | 2.64 | 54.48[65] |
| | TextVQA | 31.98 | 39.60 | **43.76** | 38.92 | 21.78 | 38.76 | 21.52 | 17.52 | 73.1[66] |
| | STVQA | 20.98 | 28.30 | **32.33** | 28.40 | 12.20 | 8.30 | 15.23 | 12.88 | - |
| | OCR-VQA | 38.85 | **60.20** | 38.12 | 23.40 | 16.15 | 3.40 | 19.50 | 16.97 | - |
| | OKVQA | 44.93 | **60.52** | 55.93 | 54.36 | 30.06 | 22.89 | 49.01 | 45.31 | - |
| | GQA | 45.53 | **49.96** | 43.93 | 41.30 | 27.03 | 12.60 | 38.12 | 38.54 | 72.1[67] |
| | Visdial | 10.73 | **45.20** | 12.92 | 14.66 | 7.97 | 13.34 | 11.67 | 12.10 | 68.92[68] |
| | IconQA | **62.82** | 56.25 | 41.83 | 42.95 | 28.20 | 09.12 | 26.77 | 25.73 | 83.62[69] |
| | VSR | 63.63 | 41.28 | 50.63 | 51.24 | 41.04 | 10.11 | 06.40 | 37.00 | 70.1[70] |
| KGID | ScienceQA IMG | **60.73** | 46.26 | 54.19 | 49.33 | 20.18 | 2.80 | 27.22 | 20.43 | 92.53[71] |
| | VizWiz | **65.44** | 65.31 | 62.07 | 62.42 | 40.76 | 11.14 | 50.04 | 11.99 | 73.3[66] |
| VE | SNLI-VE | 34.00 | 56.20 | 56.80 | **57.00** | 52.60 | 55.00 | 56.60 | 47.60 | - |
| Average Score | | 0.758 | **0.900** | 0.835 | 0.769 | 0.481 | 0.324 | 0.523 | 0.462 | - |

Table 4: Comparison of Zero-shot Performance for LVLM Models on VQA, KGID, and VE Tasks. For VQA and KGID tasks, Mean Reciprocal Rank (MRR) is used for the Visdial, while top-1 accuracy is employed for the remaining tasks.

VPGTrans achieve dominant performance in all tasks. This may be because these models use a large visual encoder (i.e., ViT-g/14) and Q-Former updated with massive image-text pairs. A stronger visual encoder and adaption module can extract better tokens entailed with the global and local context, leading to remarkable improvement in visual knowledge acquisition. Second, InstructBLIP presents consistently the best results on all tasks. The main reason is that InstructBLIP overfits these tasks by fine-tuning massive VQA data.

## 3.3 Results on Visual Reasoning

Visual reasoning encompasses the ability to comprehensively understand images and perform cognitive tasks. In this section, we evaluate the visual reasoning ability of LVLMs on various tasks, including Visual Question Answering (VQA), Knowledge-Grounded Image Description (KGID), and Visual Entailment (VE) tasks. The evaluation details of tasks are demonstrated in Appendix.B.3. Table 4 shows the zero-shot performance in visual reasoning, and we have the following observations. First, compared with BLIP2, InstructBLIP again presents better results overall because it overfits many tasks by fine-tuning massive VQA data. Second, compared with BLIP2, instruction-tuned LVLMs, except for InstructBLIP, generally perform worse than BLIP2. The common words in the instruction data often influence the generated content, which can not be evaluated by the current metrics (see Appendix C). Third, instruction-tuned LVLMs consistently surpass BLIP2 on SNLI-VE where the final answer is obtained by multi-turn reasoning. It shows that instruction-following fine-tuning can produce promising content once a good evaluation scheme is employed.

## 3.4 Results on Visual Commonsense

The visual commonsense evaluation aims to evaluate the model's comprehension of commonly shared human knowledge about generic visual concepts. We use two challenging visual commonsense

| Datasets | | BLIP2 | InstructBLIP | LA-v2 | LLaVA | MiniGPT-4 | mPLUG-Owl | Otter | VPGTrans | S-SOTA |
|---|---|---|---|---|---|---|---|---|---|---|
| ImageNetVC | Color | 44.60 | **67.79** | 23.16 | 41.92 | 26.57 | 25.56 | 26.21 | 24.72 | 44.70[15] |
| | Shape | 40.14 | **59.06** | 28.16 | 38.74 | 22.88 | 30.72 | 34.19 | 24.69 | 40.50[15] |
| | Mater. | 61.49 | 63.58 | 32.51 | **64.91** | 29.50 | 34.24 | 35.81 | 27.21 | 61.90[15] |
| | Compo. | 53.86 | **83.25** | 50.38 | 58.53 | 59.96 | 49.47 | 50.72 | 57.21 | 54.00[15] |
| | Others | 51.50 | **68.37** | 32.64 | 59.06 | 38.86 | 35.11 | 34.39 | 36.39 | 51.70[15] |
| | Avg | 50.30 | **68.41** | 33.37 | 52.63 | 35.55 | 35.02 | 36.26 | 34.04 | 50.50[15] |
| VCR | VCR | 36.80 | 45.60 | **46.20** | **46.20** | 44.40 | 39.40 | 39.60 | 39.60 | - |
| Average Score | | 0.747 | **0.994** | 0.567 | 0.807 | 0.581 | 0.564 | 0.581 | 0.546 | - |

Table 5: Comparisons of Zero-shot visual commonsense Performance for LVLM Models on VCR and ImageNetVC datasets. Top-1 accuracy is employed for the two datasets.

| Datasets | | BLIP2 | InstructBLIP | LA-V2 | LLaVA | MiniGPT-4 | mPLUG-Owl | Otter | VPGTrans | S-SOTA |
|---|---|---|---|---|---|---|---|---|---|---|
| MSCOCO | Random | 82.21 | **88.83** | 74.44 | 51.52 | 52.58 | 40.65 | 61.40 | 47.92 | - |
| | Popular | 80.10 | **84.15** | 56.82 | 50.00 | 49.31 | 38.82 | 49.56 | 47.64 | - |
| | Adversarial | 78.52 | **81.95** | 60.52 | 50.00 | 49.62 | 38.04 | 50.68 | 45.95 | - |
| Average Score | | 0.945 | **1.00** | 0.750 | 0.595 | 0.594 | 0.461 | 0.633 | 0.555 | - |

Table 6: Evaluation results of POPE [14] performance of LVLMs on MSCOCO. The accuracy is used to assess the performance.

benchmarks in a zero-shot setting, including ImageNetVC and Visual Commonsense Reasoning (VCR). The evaluation details of tasks are demonstrated in Appendix.B.4. As shown in Table 5, we can find that all those LVLMs represent their abilities to solve visual commonsense problems. First, InstructBLIP performs best (68.41%) among those LVLMs on the ImageNetVC dataset. The main reason is that it is fine-tuned on 1.6M fine-grained VQA data, making it adapt to answer visual common questions. Second, LLaMA-Adapter V2 (46.20%) and LLaVA (46.20%) show the same best performance among those LVLMs on the VCR dataset. The main reason is that instruction-flowing data is used to update the LLM. Note that the final answer of VCR is obtained by multi-turn reasoning. It also shows the significant role of a good evaluation scheme in producing promising content for instruction-tuned models.

## 3.5 Results on Object Hallucination

Although LVLMs have made significant progress, they still struggle with the issue of hallucination, which refers to their tendency to produce objects that do not align with the descriptions provided in the target images. In this section, we focus on evaluating such object hallucination problems on MSCOCO captioning dataset. Following POPE [14] evaluation pipeline which is a multi-step QA procedure, we prompt LVLMs with multiple Yes-or-No questions. For example, '*Is there a person in the image?*'. We use accuracy as the evaluation metric. From Table 6, we could come to the following conclusions. InstructBlip performs best in the hallucination problem, followed by BLIP2, whose average accuracy both reached more than 80%. We find that instruction-tuned models, except for InstructBLIP, perform worse than BLIP2 because they tend to answer 'Yes' to the question, which shows that LVLMs are prone to generate objects frequently occurring in the instruction data. Such object hallucination problem can be alleviated by a multi-turn reasoning pipeline shown in the experiments on SNLI-VE and VCR.

## 3.6 Results on Embodied Intelligence

In this section, we present the evaluation results focusing on embodied intelligence. To appraise the effectiveness of planning outputs using the given image, we conducted a user study involving 15 participants. The study comprised 6 household scenarios carefully selected from VirtualHome [51]. Specifically, the participants rated the generated plans from different LVLM models using a scoring system similar to [49]. The evaluation comprised five dimensions with scores ranging from 1 to 5. These dimensions included object recognition accuracy, spatial relationship understanding, level of conciseness in the response, reasonability of the planning, and executability of the planning. The resulting average scores for the different models among the participants are presented in Table 7 below. Furthermore, in the Appendix C, we present quantitative evaluation results for Franka Kitchen [52], Minecraft [50], and Meta-World [72]. Based on the evaluation results, we observe that visual

| Dataset | | BLIP2 | InstructBLIP | LA-V2 | LLaVA | MiniGPT-4 | mPLUG-Owl | Otter | VPGTrans |
|---|---|---|---|---|---|---|---|---|---|
| VirtualHome | Object Recon.(↑) | 2.03 | 3.08 | 3.81 | **3.88** | 3.70 | 3.42 | 3.38 | 3.43 |
| | Spatial Relation.(↑) | 1.68 | 2.78 | **3.71** | 3.61 | 3.47 | 3.22 | 3.10 | 3.22 |
| | Conciseness (↑) | **3.25** | 2.48 | 2.04 | 1.86 | 1.62 | 1.48 | 1.86 | 1.76 |
| | Reasonability(↑) | 2.78 | 3.20 | **4.04** | 3.70 | 3.54 | 3.44 | 3.07 | 3.35 |
| | Executability(↑) | 2.88 | 3.10 | **4.08** | 3.82 | 3.11 | 3.54 | 3.12 | 3.35 |
| Average Score | | 0.674 | 0.772 | **0.922** | 0.879 | 0.805 | 0.785 | 0.761 | 0.789 |

Table 7: Generated planning quality evaluation on embodied tasks. Five dimensions including object recognition, spatial relationship, conciseness, reasonability, and executability are used to assess the performance.

instruction data is essential for embodied tasks. BLIP2 lacked visual instruction tuning, which greatly affected its capability to produce reasonable and executable plans.

### 3.7 Results on Online Arena Evaluation

The arena features anonymous and randomized pairwise battles in a crowd-sourced manner. We have collected 634 pieces of evaluation data since we launch the LVLM arena. The collected data shows almost the same number of battle outcomes for 'Model A wins' and 'Model B wins.' Moreover, 21.8% battle outcomes are voted as 'both are bad,' implying that the current LVLMs still struggle to generate good answers for open-world visual questions. Furthermore, we rank the selected 8 LVLMs with Elo rating [73] using the collected data by following Fastchat [53] and [74]. As shown in Fig. 1 (b), mPLUG-Owl, MiniGPT-4, and Otter, which are fine-tuned with amounts of instruction-following data with updating many parameters, are the top-3 best models in the open-world VQA scenario, indicating the significance of instruction-following tuning and effective parameter update. Moreover, InstructBLIP perform best on in-domain capability evaluation, while being much worse than many instruction-tuned models, implying severe overfitting issue, as shown in Fig. 1.

### 3.8 Takeaway Analysis

We can conclude some actionable insights from our evaluation results. *First*, the quality of visual instruction data matters more than quantity in the open-world VQA. We observe that MiniGPT-4, which is tuned by only 3.5K high-quality visual instruction data performs much better than InstructBLIP tuned on visual instruction data adapted from various existing VQA datasets in our Multi-Modality Arena. *Second*, a strong visual encoder can help extract detailed information from the image, leading to good performance in OCR tasks. For instance, we see that BLIP2, InstructBLIP, and VPGTrans achieve better performance than the remaining 5 LVLMs. This may be because the visual encoder ViT-g/14 used in BLIP2, InstructBLIP, and VPGTrans is more powerful than ViT-L/14 employed in the remaining LVLMs. *Third*, multi-turn reasoning helps alleviate the hallucination issue, indicating that the evaluation method with critical thinking can induce the correct prediction from the model. We find that LVLM with multi-turn reasoning can determine whether an object exists in the image more accurately than single-turn reasoning. Hence, multi-turn reasoning is appropriate to assess the full potential of the model. *Fourth*, LVLMs tuned with high-quality instruction-following data present more promising planning ability than models without being tuned with instruction data as demonstrated in Table 7.

## 4 Conclusion

This paper proposes a comprehensive evaluation benchmark for large vision-language models called LVLM-eHub that incorporates both quantitative performance evaluation and human feedback evaluation. For the quantitative evaluation, we employ 16 tasks spanning over 40+ text-related visual datasets to assess the six essential capabilities of LVLM models. Additionally, we have established an online LVLM Arena to gather human feedback on LVLM models continually. This arena serves as an invaluable resource, providing an Elo rating rank that offers LVLMs ranking in the open-world scenario. Our evaluation results reveal several important findings, stimulating the future development of LVLMs. We will make ongoing efforts to build a platform for LVLM evaluation as discussed in Sec. A.4.

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
