

(a) Quantitative Capability Evaluation

| Ranking | Model | Score |
|---|---|---|
| 1 | mPLUG-Owl | 1027.0 |
| 2 | MiniGPT-4 | 1021.3 |
| 3 | Otter | 1013.2 |
| 4 | LLaMA-Adapter V2 | 1010.2 |
| 5 | LLaVA | 1009.7 |
| 6 | InstructBLIP | 1003.7 |
| 7 | VPGTrans | 974.3 |
| 8 | BLIP2 | 949.4 |

(b) LVLMs Arena Ranking

Figure A.1: Comparative analysis of LVLMs within the LVLM-eHub. (a) illustrates quantitative capability performance represented by average score in Table 2 to Table 7 in six distinct aspects among LVLMs. (b) presents the Elo rating ranking of LVLMs obtained from our LVLM Arena according to the data collected from May, 12 to June 3 in 2023.

# Appendix

In the appendix, we provide more details of LVLM-eHub and task settings of evaluation in Sec. A and Sec. B, respectively. Additionally, more experiments are illustrated in Sec. C. The evaluation datasets are summarized in Sec. D.

# A    More details about our LVLM-eHub

## A.1    Overall Evaluation Results

**Remake of Fig. 1 in the Main Text.** We apologize for the mistake in Fig. 1 where we put the radar labels of object hallucination and embodied intelligence in the wrong places. The new result is shown in Fig. A.1.

**Our Findings.** We present our observations from extensive evaluation experiments in the following.

• *Instruction-tuned LVLM with massive in-domain data such as InstructBLIP heavily overfits many existing tasks, generalizing poorly in the open-world scenario.* As shown in Fig. A.1, InstructBLIP achieves the best results in 5 categories of capabilities while lagging behind other instruction-tuned models such as LLaMA-Adapter V2 and mPLUG-Owl in embodied AI and LVLM arena platform. We see that InstructBLIP is fine-tuned on 16M visual question answering pairs (see Table 1), exhibiting superior performance in many in-domain tasks such as perception and reasoning tasks. However, Embodied AI tasks require that the model is capable of generating a step-by-step plan for an instruction with a given image. Moreover, the arena platform evaluates LVLMs' ability of visual question answering in open-world scenarios. InstructBLIP overfits in-domain tasks, generalizing poorly in these two real-world tasks.

• *Instruction-tuned LVLM with moderate high-quality instruction-following data may result in object hallucination issues.* The issue means that LVLMs would generate objects that are inconsistent with target images in the descriptions. It either makes the current evaluation metric such as CIDER for image captioning ineffective or generates wrong answers. For instance, LLaMA-Adapter V2 can generate high-quality image captions which yet present a low CIDEr score as shown in Fig. A.2. But the high sentence similarities between the generated answer and ground-truth answers measured by Sentence Transformer [75] and GPT3.5 shows that the generated answer is relatively accurate. Therefore, the instruction-tuned models could generate content that cannot be evaluated by existing metrics. It also indicates that it is urgent to develop an effective metric for LVLM evaluation.

In addition, we also find that instruction-tuned LVLMs with moderate high-quality data are more likely to generate wrong answers. As shown in Table A.2, LLaMA-Adapter V2, LLaVA, MiniGPT-4, mPLUG-Owl, Otter, and VPGTrans generally present higher accuracy and recall, and lower precision than BLIP2 and InstructBLIP. These models are tuned with moderate high-quality data such as LLaVA-158K or instruction-following data generated by LLM as shown in Table 1.

This implies that instruction-tuned LVLMs with moderate high-quality data are prone to answer 'Yes' regardless of the accuracy of the answer to the underlying question.

- *Employing a multi-turn reasoning evaluation framework can mitigate the issue of object hallucination, shedding light on developing an effective metric for LVLM evaluation.* In Table 4 and Table 5, we see that instruction-tuned LVLMs with moderate high-quality data can achieve better performance than BLIP on SNLI-VE and VCR tasks under a multi-turn reasoning evaluation pipeline in Sec. 2.3. Here we provide more evidence to demonstrate the effectiveness of such an evaluation technique in Fig.A.4. We can see that multi-turn reasoning evaluation can alleviate object hallucination issue.

## A.2    Model Details in LVLM-eHub

- **BLIP2** [6] pre-trains a lightweight Q-Former on 129M image-text pairs. It follows a two-stage strategy to bridge the modality gap. The first stage bootstraps vision-language representation learning from a frozen image encoder ViT-g/14 in EVA-CLIP [76]. The second stage bootstraps vision-to-language generative learning from a frozen LLM FlanT5-XL [77], which enables zero-shot instructed image-to-text generation.
- **LLaVA** [9] connects the visual encoder ViT-L/14 of CLIP [78] with the language decoder LLaMA [1] by a lightweight fully-connected (FC) layer. LLaVA first trains the FC layer with 595K image-text pairs while freezing the visual encoder and LLM and then fine-tunes the FC layer and LLM on 158K instructional vision-language data.
- **LLaMA-Adapter V2 (LA-V2)** [7] is a parameter-efficient visual instruction model. Although the visual encoder (ViT-L/14) and LLM are kept frozen, LLaMA-Adapter V2 distributes the instruction-following ability of the whole LLaMA through bias(B)-tuning. In this way, the scale, bias, norm, and prompt parameters are tuned on 567K image captioning data and instruction-following data.
- **MiniGPT-4** [10] connects the visual encoder and text encoder by an FC layer. It also first trains the FC layer with 5M image-text pairs and then fine-tunes it on 3.5K high-quality instructional vision-language data. Despite the simplicity, MiniGPT-4 needs to load a pretrained vision encoder of BLIP2 and Vicuna LLM [3].
- **mPLUG-Owl** [11] connects visual encoder ViT-L/14 and LLM(LLaMA) by a visual abstractor, which is instantiated by a cross-attention module with several learnable query tokes. During pre-training, mPLUG-Owl trained the visual encoder and visual abstractor on 204M image-text pairs. For the second stage, 158K LLaVA-Instruct data is utilized to train the LoRA weights of LLaMA.
- **Otter** [12] is a multimodal model with in-context instruction tuning based on OpenFlamingo [5] which comprises a LLaMA-7B language encoder and a CLIP ViT-L/14. Although the visual and text encoder are frozen, Otter trains extra 1.3B parameters coming from adaption modules on 158K instruction-following data.
- **InstructBLIP** [13] is initialized from a pre-trained BLIP-2 model consisting of a ViT-g/14 image encoder, a Vicuna LLM, and a Q-Former to bridge the two. During vision-language instruction tuning, only Q-Former is fine-tuned on 13 visual question-answering datasets.
- **VPGTrans** [8] is a simple transferring technique that adapts a smaller LLM to a larger LLM. It transfers the VPG of BLIP-2 (i.e. ViT-g/14) from OPT6.7B to Vicuna7B by training Q-Former on 13.8M Image-Text pairs. In addition, the VPG and projector are further tuned on MiniGPT-4's 3.5K self-instruct data instances.

## A.3    Evaluation Metrics

Note that the concrete evaluation metrics for each dataset are provided in the caption of Table 2 to Table 7. For evaluation methods, we use the word matching technique in a question-answering fashion to assess the performance of LVLMS for all benchmarks except that ImageNetVC is evaluated by the prefix-based score and SNLI-VE and VCR are evaluated by multi-turn reasoning. We present a detailed formulation of the prefix-based score and multi-turn reasoning as follows.

**Formulation of prefix-based score.** Prefix-based Score method treats the zero-shot evaluation as a cloze test using prompts. QA pairs are transformed into prompts like "[Question] The answer is [Answer]". Each QA pair is converted into a sequence of tokens $x = \{x_0, ..., y, ..., x_n\}$ via a prompt, in which $y$ is one of the candidate answers. Those tokens will first be mapped to text embeddings $e_t = \{e_t(x_0), ..., e_t(y), ..., e_t(x_n)\}$ by the embedding layer. Then we use a visual encoder to transformer the image $v$ of a QA pair into a sequence of visual embeddings $e_v = \{e_v^1, ..., e_v^m\}$. Next, the visual embeddings are prefixed into the text embeddings and then put into the LLM backbone to

calculate the score for the answer $y$. The detailed formulation is as followed:

$$s(y|v,x) = \frac{1}{|y|} \sum_{i=1}^{|y|} \log P(y_i|[e_v, e_t]). \tag{1}$$

where $P(y|[e_v, e_t])$ is the probability of generating the candidate answer which is obtained by running a forward of the LLM and $|y|$ is the token length of the answer. We treat $s(y|v,x)$ as the log-likelihood of producing answer $y$.

Finally, a probability distribution over all answer candidates using softmax is given by:

$$q(y|v,x) = \frac{e^{s(y|v,x)}}{\sum_{y' \in \mathbf{y}} e^{s(y'|v,x')}} \tag{2}$$

where $\mathbf{y}$ denotes the set of candidate answers. We choose the answer with the highest probability as the correct answer.

**Formulation of Multi-turn reasoning.** Multi-turn reasoning method iteratively decomposes vision and language reasoning with large language models. There are three components in the framework: a Questioner (CharGPT), an Answerer (LVLM), and a Reasoner (ChatGPT). Given a main question $q$ and an image $I$, and answer candidates $A = \{a_1, ..., a_n\}$, The Questioner needs decompose the main question into several sub-questions $SubQ = \{sq_1, .., sq_i\}$. Answerer then provides the corresponding answers and finally, a Reasoner reasons to achieve the final answer.

Specifically, we design a prompt $P_q$ as an instruction to generate sub-questions by ChatGPT. To enable the Questioner to understand the image and generate more informative questions, we also use the underlying LVLM to obtain a caption $C$. In the first iteration, the main question $q$, the prompt $P_q$ and the caption input into ChatGPT to obtain the sub-questions. The first iteration process can be formulated as follows:

$$SubQ_1 = ChatGPT(q, C, P_q). \tag{3}$$

Subsequently, we loop back to the Questioner to generate additional supplementary informative sub-questions to obtain more sufficient evidence. In the t-th iteration (t>1), Questioner accepts all previous sub-questions $SubQ_{1:t-1}$ and sub-answers $SubA_{1:t-1}$, and the previous analysis from Reasoner $E_{t-1}$ as additional input. The following iteration processes can be computed as follows:

$$SubQ_t = ChatGPT(q, C, P_q, SubQ_{1:t-1}, SubA_{1:t-1}, E_{t-1}). \tag{4}$$

Where $SubQ_{1:t-1} = \{SubQ_1 \cup .... \cup SubQ_{t-1}\}$ and $SubA_{1:t-1} = \{SubA_1 \cup .... \cup SubA_{t-1}\}$. Previous sub-questions and sub-answers can inform Questioner what has been asked and solved, and the analysis can guide Questioner to generate more specific sub-questions.

As for the Answerer, given the generated sub-questions SubQ, Answerer is used to answer them correspondingly to provide evidence for answering the main question. We use the underlying LVLM studied in this paper to answer each sub-question separately:

$$sa_i = LVM(sq_i, I). \tag{5}$$

Where $sq_i \in SubQ$ and $sa_i \in SubA$.

As for the Reasoner, we use ChatGPT to analyze both SubA and SubQ to decide if a confident answer $a$ to the main question $q$ can be derived. The main question $q$, caption $C$, all existing sub-questions $SubQ_{1:t}$ and corresponding sub-answers $SubA_{1:t}$ are fed into the Reasoner. The Reasoner is prompted to generate both the analysis and the final answer with its prompt $P_R$.

$$E_t, a = ChatGPT(SubQ_{1:t}, SubA_{1:t}, q, C, P_R). \tag{6}$$

If the Reasoner is not confident about the final answer, it is instructed to faithfully indicate that by generating a specific response such as "We are not sure". If this particular response is detected, we start another iteration by asking the Questioner to add supplementary sub-questions. The above procedure forms a loop among the three agents, which will stop if the Reasoner can find confident answer or the number of iterations reaches a pre-defined bound.

| | CIDEr Score | Sim. | GPT3.5 turbo |
|---|---|---|---|
| The image features a close-up view of a small, sparkling tiara. | | | |
| A button with a tiny encrusted tiara on it. | | 0.613 | 9/5 |
| The studded crown sits on the turn table. | | 0.241 | 8/6 |
| A tiara is sitting on a grey surface. | 0.001 | 0.555 | 7/6 |
| Tiara with heart shaped pattern on black platform. | | 0.654 | 8/5 |
| A silver crown sitting on top of something plastic and black. | | 0.318 | 7/5 |
| The image shows a close-up view of a large, messy, and delicious-looking cheeseburger. | | | |
| Hamburger with cheese and bacon from fast food. | | 0.448 | 9/8 |
| This large cheese burger has bacon on it. | | 0.575 | 9/8 |
| A hand holding a big bacon and cheese hamburger. | 0.001 | 0.454 | 8/4 |
| A person is holding a sandwich with his/her hand. | | 0.345 | 5/2 |
| A person holding a hamburger with bacon on it. | | 0.361 | 8/5 |

Figure A.2: **Limitations of CIDEr Score Evaluation in Image Captioning**. SentenceTransformer [75] computes the similarity between generated and ground-truth text as "Sim." "GPT3.5 Turbo" indicates that we feed GPT with the most elaborate ground-truth text and use it to evaluate the quality of the generated text (left score) and other ground-truth texts (right score). The template is similar to Vicuna's GPT-4 evaluation [3] but replaces the question with ground-truth text.

## A.4 A Platform for LVLM Evaluation.

We have developed an evaluation framework aimed at comprehensively assessing the performance of LVLM models across six critical capabilities. Each capability encompasses multiple tasks, with several datasets incorporated into each task. Our user-friendly interface allows users to contribute their own datasets and models, facilitating a collaborative and inclusive environment. With just one click, users can effortlessly access a holistic assessment of their target LVLM model through our evaluation platform. We are dedicated to regularly updating the datasets and expanding our support for a wider range of LVLM models on our platform. Users are encouraged to contribute their LVLM models by utilizing our platform's model inference interface. Additionally, we offer free online inference services for the LVLM models supported by LVLM Arena. This arena not only allows users to vote for their preferred models but also provides an Elo rating ranking system that incorporates valuable human feedback, ensuring continuous improvement and refinement.

# B Evaluation Details

## B.1 Details of Visual Perception

**For ImgCls**, we test LVLMs on two coarse-grained benchmarks (*i.e.,* ImageNet1K and CIFAR10) and two fine-grained benchmarks (*i.e.,* Pets37 and Flowers102). Following KOSMOS-1 [79], the default prompt '*The photo of the*' is used for all LVLMs for a fair comparison. However, the generated coherent sentence-style responses deviate from the standard image classification benchmark. To accommodate this discrepancy, we considered the prediction as correct if the model output contains the correct class name, which is inspired by MultiModal OCR[16].

**For OC** task, we test LVLMs on MSCOCO and VCR1.0 [48]. It involves querying the model about the number of objects belonging to an image's specific class of interest. To this end, we use the prompt '*Question: How many [obj] are there in the image? Answer:*'. The generated answer is then compared with the ground truth. We report accuracy by treating OC as a classification problem.

**For MCI** task, we also test LVLMs on MSCOCO and VCR1.0 [48]. We ask the model to determine whether a certain object is present or absent by prompting '*Question: Does [obj] exist in the image. Answer:*'. We also report the accuracy by treating MC as a Yes or No classification problem.

## B.2 Details of Visual Knowledge Acquisition

**For OCR** task, we test the selected LVLMs with twelve representative OCR datasets, which are inclusive of IIIT5K[30], ICDAR 2013(IC13)[31], ICDAR 2015 (IC15)[32], Total-Text[33], CUTE80[34], Street View Text (SVT)[35], SVTP-Perspective (SVTP)[36], COCO-Text[37], WordArt[38], SCUT-

CTW1500 (CTW)[39], heavily occluded scene text (HOST)[40], weakly occluded scene text (WOST)[40]. These benchmarks consist of a diverse range of images containing textual information which can make an adequate comparison between LVLMs. The performance of the LVLMs is compared with top-1 accuracy and the prompt we use is '*what is written in the image?*'.

**For KIE** task, we employ the SROIE[41] and FUNSD[42] benchmarks to evaluate LVLMs, which encompass diverse documents like receipts and forms that require specific information extraction. The performance of LVLMs is evaluated using entity-level F1 scores. Additionally, we utilize information-specific prompts for each piece of information that the model should extract. For instance, in the SROIE benchmark case, we use the prompt '*what is the name of the company that issued this invoice?*' to extract company information and '*where was this invoice issued?*' prompt for address information. Please refer to the Appendix for more detailed information.

**For ImgCap** task, we utilize two benchmarks, including NoCaps[43] and Flickr30K[44]. Each benchmark provides a collection of images with corresponding captions. In evaluation, CIDEr scores are used to evaluate these models with the prompt '*what is described in the image?*'.

### B.3   Details of Visual Reasoning

**For VQA** task, we utilize nine benchmarks: DocVQA[65], TextVQA[80], STVQA[81], OCR-VQA[82], OKVQA[83], GQA[84], IconQA[69], Visual Spatial Reasoning (VSR)[70], and Visual Dialog (Visdial). These benchmarks offer a diverse set of question-image pairs, covering a wide range of topics. The task requires LVLMs to not only understand the visual content but also comprehend and reason about the posed questions. For specific evaluation, we employ the Mean Reciprocal Rank (MRR) metric for Visdial and top-1 accuracy for the remaining datasets. These metrics provide insights into the model's ability to accurately answer questions across the various VQA benchmarks.

**For KGID** task, it evaluates the LVLM's capability to generate informative and accurate descriptions of images by incorporating external knowledge. To assess performance, we employ the ScienceQA[46] and VizWiz[47] benchmarks, which consist of images accompanied by textual descriptions and knowledge-based information. Notably, in the case of ScienceQA, we only utilize the samples that contain images.

**For VE** task, it evaluates the VLPM's capability to determine the logical relationship between image pairs. We employ the SNLI-VE [26] benchmark, which provides pairs of images along with corresponding textual premises and hypotheses. We find that a naive QA pipeline is hard to give meaningful predictions. We thus employ multi-turn reasoning to solve SNLI-VE.

### B.4   Details of Visual Commonsense

**For ImageNetVC**, we evaluate the zero-shot visual commonsense of LVLMs. It contains high-quality QA pairs for various commonsense, including color, shape, mater, comp, and others. For a QA pair, we use the text prompt '*[Question] The answer is [Answer].*'. We then use a prefix-based score to choose the final answer with the maximum likelihood.

**For VCR**, it expects that the LVLMs can find the correct answer among four answer candidates. For efficient evaluation, we randomly select 500 samples from the val split of the VCR dataset. We find that a naive QA prompt cannot give meaningful output. Similar to the SNLI-VE evaluation, we adopt a multi-turn reasoning evaluation technique to solve the VCR task.

## C   More Experiments

### C.1   Sensitivity to Prompts.

Throughout our comprehensive evaluation, we discovered that LVLM models are highly sensitive to the choice of prompts. An illustrative example of this sensitivity is observed in the image captioning task, where altering the prompt employed for the VPGTrans model on the NoCaps dataset leads to a substantial variation in performance, ranging from 19.66 to 36.20. Fig. A.3 showcases some examples of the results generated based on different prompts. This sensitivity underscores the significance of carefully selecting and designing prompts to achieve optimal performance in LVLM-based tasks.

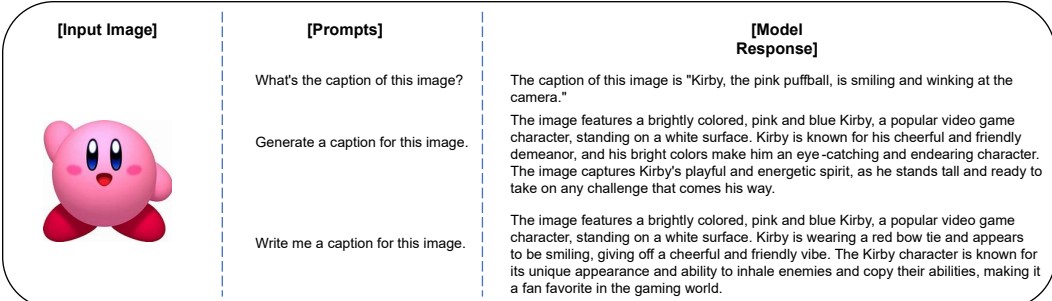

| [Input Image] | [Prompts] | [Model Response] |
|---|---|---|
| | What's the caption of this image? | The caption of this image is "Kirby, the pink puffball, is smiling and winking at the camera." |
| | Generate a caption for this image. | The image features a brightly colored, pink and blue Kirby, a popular video game character, standing on a white surface. Kirby is known for his cheerful and friendly demeanor, and his bright colors make him an eye-catching and endearing character. The image captures Kirby's playful and energetic spirit, as he stands tall and ready to take on any challenge that comes his way. |
| | Write me a caption for this image. | The image features a brightly colored, pink and blue Kirby, a popular video game character, standing on a white surface. Kirby is wearing a red bow tie and appears to be smiling, giving off a cheerful and friendly vibe. The Kirby character is known for its unique appearance and ability to inhale enemies and copy their abilities, making it a fan favorite in the gaming world. |

Figure A.3: In this example, we present the variation in model responses of LA-V2 when provided with the same image but different prompts. All model responses mentioned here were generated using zero temperature and a top-p value of 0.75. The purpose is to showcase how the model's output changes based on the prompt input.

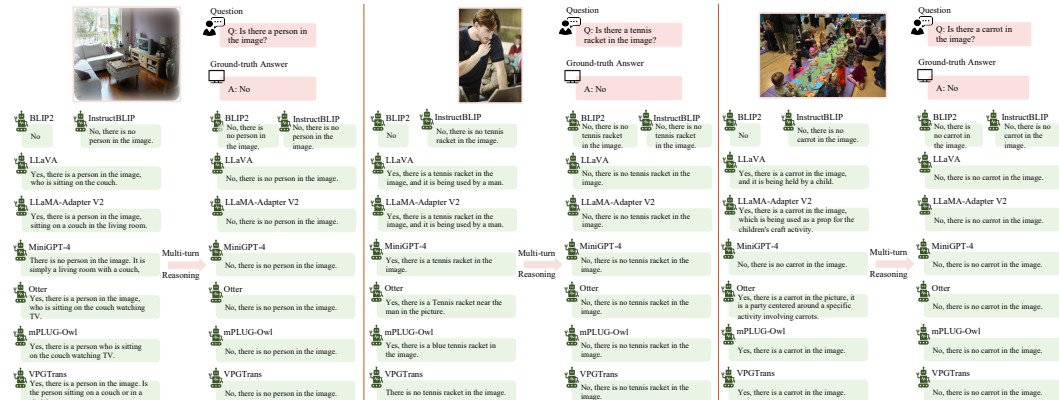

Figure A.4: **The Effect of Multi-turn Reasoning Evaluation Pipeline**. We see that many LVLMs could generate content that does not exist in the given image, suffering from the object hallucination issue [14]. Moreover, a multi-turn reasoning evaluation pipeline can mitigate object hallucination issues.

**Results of 80 prompts on image classification.** To thoroughly explore the potential impact of the prompts we utilized on the performance of VLMs, we conducted an empirical study. This investigation aimed to provide insights into the performance of VLM models across a range of prompts. We employed a set of 80 prompts, previously utilized in the prompt engineering experiment of CLIP[1], and applied them to a subset of the ImageNet1K validation set. Within this subset, we randomly selected 3 images for each class, resulting in a total of 3,000 images. This comprehensive approach enabled us to analyze the nuanced performance variations of VLMs under diverse prompts. The detailed performance metrics are presented in Table A.1. The term "baseline" refers to the accuracy of the prompt we used in our paper, while "mean" and "std" represent the statistical performance measures derived from the aforementioned set of 80 prompts. We can see that BLIP2 and InstructBLIP still achieve competitive performance on the ImageNet validation subset over 80 prompts. However, the results averaged over 80 prompts present moderate variations.

## C.2    More Results on Object Hallucination

**Object Hallucination Measured by More Metrics.** In the main text, we evaluate LVLMs in object hallucination issues by the accuracy in answering a series of Yes-or-NO questions. Here we complete it with more evaluation metrics such as precision and recall as shown in Table A.2. From Table A.2, we can see that instruction-tuned models with moderate data such as LLaMA-Adapter V2 and LLaVA

---

[1]https://github.com/openAI/CLIP/blob/main/notebooks/Prompt_Engineering_for_ImageNet.ipynb

| | BLIP2 | InstructBLIP | LLaMA-Adapter-v2 | LLaVA | MiniGPT-4 | mPLUG-Owl | Otter | VPGTrans |
|---|---|---|---|---|---|---|---|---|
| Main Text | 24.1 | 24.4 | 19.2 | 18.2 | 16.4 | 20.4 | 14.5 | 17.1 |
| Mean (80 prompts) | 26.5 | 24.3 | 18.3 | 17.5 | 13.8 | 18.2 | 13.8 | 13.2 |
| Std (80 prompts) | 3.28 | 2.47 | 1.35 | 1.38 | 2.31 | 2.74 | 0.73 | 1.30 |

Table A.1: The Performance of LVLMs on ImageNet subset (3000 images) under 80 different prompts.

| Datasets | Metrics | BLIP2 | InstructBLIP | LA-V2 | LLaVA | MiniGPT-4 | mPLUG-Owl | Otter | VPGTrans |
|---|---|---|---|---|---|---|---|---|---|
| MSCOCO-Random | Accuracy | 82.21 | **88.83** | 74.44 | 51.52 | 52.58 | 40.65 | 61.40 | 47.92 |
| | Precision | 97.48 | 96.01 | 68.24 | 51.54 | 68.63 | 62.96 | 57.82 | 71.70 |
| | Recall | 67.27 | 81.60 | 94.00 | 100.00 | 57.50 | 97.45 | 95.92 | 56.07 |
| | F1-Score | 79.61 | **88.23** | 79.08 | 68.03 | 62.57 | 76.50 | 72.15 | 62.93 |
| | Yes | 35.58 | 43.99 | 70.99 | 100.00 | 44.25 | 35.37 | 85.76 | 47.68 |
| MSCOCO-Popular | Accuracy | 80.10 | **84.15** | 56.82 | 50.00 | 49.31 | 38.82 | 49.56 | 47.64 |
| | Precision | 90.49 | 85.96 | 53.89 | 50.00 | 63.56 | 56.57 | 50.07 | 70.37 |
| | Recall | 67.27 | 81.60 | 94.20 | 100.00 | 58.03 | 97.13 | 95.92 | 55.60 |
| | F1-Score | 77.17 | **83.72** | 68.56 | 66.67 | 60.67 | 71.50 | 65.79 | 62.11 |
| | Yes | 37.17 | 47.47 | 87.40 | 100.00 | 48.29 | 92.76 | 96.58 | 50.44 |
| MSCOCO-Adversarial | Accuracy | 78.52 | **81.95** | 60.52 | 50.00 | 49.62 | 38.04 | 50.68 | 45.95 |
| | Precision | 86.83 | 82.05 | 54.58 | 50.00 | 62.55 | 57.18 | 50.56 | 68.63 |
| | Recall | 67.27 | 81.60 | 96.45 | 100.00 | 58.71 | 97.50 | 95.92 | 56.46 |
| | F1-Score | 75.81 | **81.82** | 69.12 | 66.67 | 68.47 | 72.09 | 66.22 | 61.96 |
| | Yes | 38.73 | 49.77 | 88.23 | 100.00 | 48.54 | 94.33 | 95.31 | 51.20 |

Table A.2: Detailed evaluation results of the zero-shot performance of LVLMs on MSCOCO using POPE evaluation pipeline [14], where accuracy represents the accuracy of prediction; precision represents how many of the predicted positive samples are true positive samples; recall represents how many of all true positive samples are correctly identified; and yes represents the probability that the model outputs a yes answer.

are prone to answer 'Yes'. But they achieve lower accuracy than BLIP2 and InstructBLIP, implying that they would generate many objects that do not exist in the image, resulting in object hallucination problems.

**Multi-turn Reasoning Evaluation Can Alleviate Object Hallucination.** We also show the hallucination problem by visualizing some examples in Fig. A.4. It can be seen that LVLMs except for BLIP2 and InstructBLIP are prone to generate objects which are inconsistent with the image. Fortunately, such an issue can be mitigated by a multi-turn reasoning evaluation framework. We believe that the reasoning procedure can encourage instruction-tuned models to re-organize the knowledge they grasp and finally generate the right answers. It is significant to investigate how to evaluate instruction-tuned LVLMs in the right way. To further verify this, we directly evaluate the performance of LVLMs on COCO Random using multi-turn reasoning. For quick verification, we sample 50 samples from COCO Random. The results are reported in Table A.4. We can see that the performances of LVLMs improve a lot under multi-turn reasoning, indicating that multi-turn reasoning can alleviate the issue of object hallucination.

## C.3  More Ablation Study

**A new evaluation metrics.** Our quantitative evaluation mainly uses the CIDEr score and accuracy. The CIDEr score measures the similarities between generated and ground-truth answers. However, LVLMs' responses are diverse, in different styles with the ground truth. As such, the CIDEr score is unsuitable (see Appendix C for failure cases). We also tried model-based evaluation, which uses Sentence Transformer to calculate the feature similarity between the generated and the ground-truth answer. It is generally more robust but still suffers in some cases due to model limitations. Recent studies use the powerful Chat-GPT or GPT-4 as a judge to evaluate LLMs' responses. However, it is blind to the image and is inaccurate in vision-language tasks. We introduced LVLM Arena, a novel evaluation framework using a 1v1 LVLM battle with human judgment. However, it requires significant effort to produce reliable rating results, particularly when numerous models exist. Therefore, developing fast, accurate, and generalized evaluation metrics for LVLMs remains an open problem.

| Datasets | VLM Baseline | | | LVLM | | | | | | | |
| | CLIP | BLIP | XVLM | BLIP2 | InstructBLIP | LLaMA-Adapter-v2 | LLaVA | MiniGPT-4 | mPLUG-Owl | Otter | VPGTrans |
|---|---|---|---|---|---|---|---|---|---|---|---|
| VG-R | 51 | 57 | 65 | 45 | 73 | 60 | 42 | 63 | 37 | 5 | 47 |
| VG-A | 55 | 83 | 89 | 96 | 83 | 70 | 47 | 63 | 64 | 45 | 69 |
| COCO-O | 50 | 37 | 35 | 21 | 31 | 47 | 60 | 42 | 11 | 12 | 33 |
| F30K-O | 62 | 44 | 56 | 45 | 44 | 55 | 31 | 27 | 9 | 16 | 20 |
| Avg. Score | 54.5 | 55.3 | 61.3 | 51.8 | 57.8 | 58.0 | 45.0 | 48.8 | 30.3 | 19.5 | 42.3 |

Table A.3: The Performance of LVLMs on the ARO benchmark, which assesses the understanding of relation (VG-R split), attribution (VG-A split), and order (COCO-O and F30K-O split) in the image. The average score is obtained by averaging all results within each column.

| | BLIP2 | InstructBLIP | LLaMA-Adapter-v2 | LLaVA | MiniGPT-4 | mPLUG-Owl | Otter | VPGTrans |
|---|---|---|---|---|---|---|---|---|
| Multi-turn | 72 | 80 | 72 | 58 | 56 | 48 | 60 | 56 |
| Single-turn | 80 | 80 | 76 | 74 | 66 | 80 | 68 | 64 |

Table A.4: The performance of LVLMs on COCO Random evaluated by multi-turn reasoning. Multi-turn reasoning can alleviate the issue of object hallucination.

**The performance on Compositionality of LVLMs.** We evaluate the performance of eight models in the ARO benchmark [85], which includes VG-Relation, VG-Attribution, COCO-Order, and Flickr30k-Order. In each subset of the ARO benchmark, we randomly select 100 samples with seed 0. Different from VLMs such as BLIP [86] and CLIP, which predict the answer by comparing the similarities between the image feature and text features, LVLM obtains the final answer by transferring the ARO benchmark into a multi-choice visual question-answering task. For example, given an image showing that the horse is eating the grass, we give the LVLM model two choices: A) the horse is eating the grass, and B) the grass is eating the horse. We report the final accuracy in Table A.3. We can see that existing LVLMs exhibit intriguing deficiencies in understanding compositionality. Only Instruct BLIP, LLaMA-Adapter V2, and MiniGPT-4 exceed the chance-level accuracy on all benchmarks. There are even no LVLMs outperforming VLM model X-VLM in terms of the average score on all ARO benchmarks. Hence, it has plenty of room to improve the comprehension of LVLMs in compositionality such as relation, attribute, and order.

**Robustness to randomness in text generation.** The sampling temperatures used in our quantitative evaluation are not zero for every model. Specifically, we use the default sampling parameter used in each model's GitHub repository, as such a parameter is usually a good choice for the underlying model tuned by the model provider. To further test the sensitivity of LVLMs to randomness, we run experiments on the ImageNet validation subset (i.e. 3 images for each class and 3k images in total) three times with the same inference configuration. We can see that all LVLMs present a small accuracy variation as shown in Table A.5. This may be because the population accuracy on 3k images of ImageNet can converge despite the randomness of the single testing sample. We also verify this finding in longer-form generation tasks such as image captioning on Nocaps [43]. A similar result can also be observed on longer-form generation tasks as shown in Table A.5.

## C.4 More Results on Embodied Tasks.

In this section, we provide quantitative evaluation results for embodied tasks in addition to the user study discussed in Section 3.6. We selected some representative scenes from Minecraft, Franka Kitchen, and Meta-World benchmarks as shown in Figures A.5 through A.7, and the results for these tasks are provided in Sections C.4.1 to C.4.3.

In Figure A.5, the models were asked to generate feasible plans for the Minecraft agent to reach the opposite shore with a boat floating on the river. All eight models recognized the presence of the floating boat, but only LLaMA-Adapter V2, InstructBLIP, and MiniGPT-4 generated a plan that utilized the boat to help the agent reach the opposite shore more quickly.

In Figure A.6, the models needed to assist the robotic arm in moving the kettle to the top left burner, and we expected the models to analyze where the goal state was achieved from the image. Except for BLIP, all seven models provided a feasible and reasonable plan. Notably, LLaVA recognized that the goal state had already been achieved from the given image. Meanwhile, mPLUG-Owl generated some steps to deal with the situation when the goal state was achieved before execution.

|  | BLIP2 | InstructBLIP | LLaMA-Adapter-v2 | LLaVA | MiniGPT-4 | mPLUG-Owl | Otter | VPGTrans |
|---|---|---|---|---|---|---|---|---|
| *classification on ImageNet subset* | | | | | | | | |
| Mean (3 rounds) | 23.9 | 24.4 | 21.9 | 18.6 | 16.9 | 20.4 | 14.5 | 15.9 |
| Std (3 rounds) | 0.01 | 0.01 | 0.20 | 0.30 | 0.46 | 0.55 | 0.01 | 0.42 |
| *image captioning on Nocaps* | | | | | | | | |
| Main Text | 48.6 | 46.6 | 33.7 | 1.6 | 5.8 | 0.3 | 11.6 | 36.2 |
| Mean (3 rounds) | 48.8 | 46.2 | 33.8 | 1.2 | 6.4 | 0.2 | 11.7 | 36.6 |
| Std (3 rounds) | 0.01 | 0.01 | 0.14 | 0.11 | 1.13 | 0.04 | 0.01 | 0.56 |

Table A.5: The Performance of LVLMs on ImageNet subset (3000 images) and Nocaps. We conducted two experiments in three rounds with the same inference configurations.

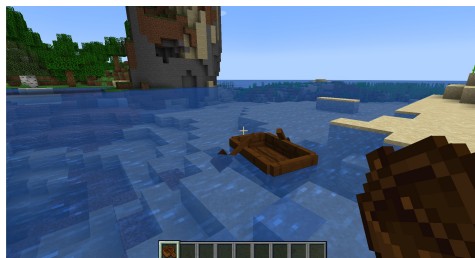

Figure A.5: Embodied image caption and plan generation example task in Minecraft simulator.

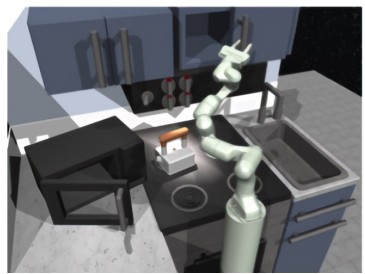

Figure A.6: Embodied image caption and plan generation example task in Franka Kitchen benchmark.

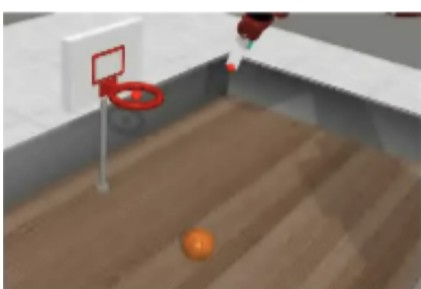

Figure A.7: Embodied image caption and plan generation example task in Meta-World benchmark.

For the Meta-World examples shown in Figure A.7, some models struggled with visual reasoning partially due to the low-resolution image input. mPLUG-Owl detected the basketball in the robot's hand mistakenly, while others, e.g. LLaVA, MiniGPT-4, and Otter, mistook the robotic arm for a human.

 **C.4.1 Minecraft**

**BLIP:** This is a Minecraft screenshot of a boat in the water.

**InstructBLIP:** The image depicts a Minecraft scene where the player is standing on a small island in the middle of a large body of water. The player's goal is to reach the opposite shore as quickly as possible, which involves navigating through the water and possibly encountering various obstacles along the way. In the scene, there are several objects visible, including a boat, a tree, and a rock. **The boat appears to be partially submerged in the water, suggesting that the player may need to navigate around it or use it as a means of transportation to reach the opposite shore.** The tree serves as a landmark, providing a reference point for the player to orient themselves and determine their progress towards the opposite shore. The rock is also present, adding to the visual interest of the scene. The spatial relationship between the player, the boat, the tree, and the rock can be inferred from the image. The player is located on the small island in the middle of the water, while the boat is partially submerged nearby. The tree serves as a reference point, indicating the player's proximity to the opposite shore. The rock adds to the visual appeal of the scene, contributing to the overall sense of.

**LLaMA-Adapter V2:** The image shows a Minecraft scene with a small wooden boat floating on a body of water. The boat is positioned near the shore, and there is a path leading to the opposite shore. The boat appears to be in a relatively calm body of water, and it seems to be the primary mode of transportation for the player. **The player's goal is to reach the opposite shore as fast as possible, so they will likely need to use the boat to navigate the water and reach their destination.**

**LLaVA:** The image shows a Minecraft scene with a small boat floating in a body of water, surrounded by a sandy beach. There is also a small island in the background. The boat is positioned near the shore, and the player is standing on the beach, looking at the boat. The player's objective is to reach the opposite shore as fast as possible. The scene appears to be a part of a game, and the player's actions are likely to be controlled by the game's mechanics.

**MiniGPT-4:** The image shows a small wooden boat floating in the water. The boat has a small cabin at the front and a mast with a sail at the rear. There is a rocky shore on one side of the boat and a sandy beach on the other. The water is calm and there are no other objects or people in the scene. The sun is shining from the top right corner of the image, casting a bright light on the water and the boat. The boat is positioned in the middle of the water, with the mast facing towards the shore. **The goal is to reach the opposite shore as fast as possible, so the player would need to navigate the boat through the water, possibly using a paddle or oars, and avoid any obstacles or hazards in the way.**

**mPLUG-Owl:** In the image, a person, likely the user, is standing in the shallows near the water, looking for a way to cross over to a boat that is on the other side of the water. They appear to be in a desert biome, as evidenced by the surrounding landscape. The boat is located at a distance from the shore, and it seems to be partially underwater, suggesting that it could be a partially submerged boat. In addition to the person standing near the water, there are several rocks and trees scattered across the scene. One of the rocks is located closer to the shore, while the others are scattered farther away from it. The trees are also dispersed throughout, some closer to the shore and others further away. The presence of these objects and their spatial relationships suggest that the scene might be part of a desert-themed map or world in Minecraft.

**Otter:** In this image, a person is standing on a raft in the middle of a body of water, with a small boat nearby on the shore. The scene is set in a game like Minecraft, where the player is trying to reach the other side of the water. The raft and the small boat are the primary objects that the player needs to utilize to cross the water safely. **The person on the raft is standing near the boat, likely preparing to use it** to continue their journey to the other shore.

**VPGTrans:** The image shows a small wooden boat floating in the middle of a large body of water. There is a small island on the other side of the water, and the boat is headed towards it. The water is calm, and there are no visible obstacles in the way of the boat. The boat appears to be in good condition, and there are no visible signs of damage or wear and tear. The island appears to be relatively close, and it should be possible to reach it in a short amount of time.

 **C.4.2   Franka Kitchen**

**BLIP:** Move the kettle to the top left burner step.

**InstructBLIP:** Here's a step plan: 1. Analyze the image to determine whether the action goal of moving the kettle to the top left burner has already been achieved or not. 2. If the action goal is not achieved, generate a plan for the robot to move the kettle to the top left burner. 3. Determine the initial position of the kettle on the stove. 4. Calculate the distance between the current position of the kettle and the desired position on the top left burner. 5. Based on the calculated distance, determine the necessary steps for the robot to move the kettle from its current position to the desired position on the top left burner. 6. Implement the necessary steps using the robot's movement capabilities. 7. Verify that the kettle has been successfully moved to the top left burner.

**LLaMA-Adapter V2:** To move the kettle to the top left burner, the robot would need to follow these steps: 1. Detect the kettle's position: The robot should first identify the kettle's current location on the kitchen counter. This can be done by analyzing the image and recognizing the kettle's shape, size, and color.2. Plan the path: The robot should then calculate the most efficient path to move the kettle to the top left burner. This could involve moving the kettle to a specific location on the counter, then rotating or tilting the kettle to align it with the burner.3. Execute the plan: The robot should then execute the plan.

**LLaVA:**  To move the kettle to the top left burner, the robot should follow these steps: 1. Identify the kettle's current position on the stove. 2. Determine the appropriate path to reach the top left burner. 3. Approach the kettle from the side, ensuring it doesn't obstruct the path. Move the kettle towards the top left burner, making sure it doesn't touch any other burners or appliances. 5. Once the kettle is positioned on the top left burner, confirm that it is stable and secure. 6. If the kettle is not yet on the top left burner, repeat steps 2-5 until the kettle is successfully placed on the burner. **In the image, the kettle is already on the top left burner, so there is no need to generate the plans. The robot has successfully moved the kettle to the desired location.**

**MiniGPT-4:**  The robot is standing in the middle of a kitchen, facing a counter with two burners on top and a sink on the left. A kettle is on the counter in front of the right burner. The robot is holding a ladle in its right arm and is reaching towards the kettle with it. Here's a plan to move the kettle to the top left burner: 1. The robot will take the ladle from its right arm and place it on the counter next to the kettle. 2. The robot will use its left arm to pick up the kettle and place it on the top left burner. 3. The robot will then take the ladle and use it to pour hot water into the kettle. Overall, the action goal is achieved in three steps.

**mPLUG-Owl:**  1. Obtain the position of the kettle on the countertop from the image. If the position of the kettle is not known, move the robot arm to scan the entire countertop area to find it. 3. After locating the kettle, the robot arm should grasp it using its gripper. 4. Using sensors such as cameras, the robot arm should navigate around other objects on the countertop and reach the desired location, which in this case is the left burner of a kitchen stove. Once the robot has arrived at the desired location, it should securely place the kettle on the top left burner. 6. The robot may need to adjust its grip on the kettle or the stove if necessary to maintain stability. 7. After the kettle is securely placed, the robot arm should release its grip and return to its original position or move back to the countertop to perform any additional tasks. Note: If it is already known that the kettle is on the top left burner, the robot may skip to step 4 or proceed to step 6 directly, based on its predefined task.

**Otter:**  In the image, the robot is already in the process of lifting the teapot with its arm and is positioning it on top of a burner on the stove. The plan would likely consist of the following steps: 1. Position the arm with the pot securely and safely on the burner. 2. Ensure that the pot is placed at the correct height and distance from the flame to avoid any accidents or damage to the pot. 3. Utilize the robot's vision system or other sensors to accurately position the pot at the desired location. 4. Once positioned, the arm should maintain a firm grip on the pot to prevent it from falling or shifting during the cooking process. 5. If the pot needs to be lowered or moved to a different burner, the same process would be repeated, with the robot carefully adjusting the position of the pot and maintaining a secure grip. The generated plan should be clear, step-by-step, and easy to follow, ensuring that the robot successfully places the pot on the desired burner while maintaining safety and efficiency.

**VPGTrans:**  To move the kettle to the top left burner, the robot should first identify the location of the kettle and the location of the top left burner. Then, the robot should use its arm to pick up the kettle and move it to the top left burner. Once the kettle is in place, the robot should use its arm to adjust the position of the kettle so that it is centered on the burner. Finally, the robot should turn on the burner to heat the water in the kettle.

 **C.4.3 Meta World**

**BLIP:** A robot is dunking a basketball in a pool.

**InstructBLIP:** The image depicts a 3D basketball game where a robot is attempting to dunk the ball. The scene features a basketball hoop, a basketball, and a robot. In order to successfully dunk the basketball, the robot needs to perform several actions. Here is a detailed plan for the robot to dunk the basketball: 1. Position the robot near the basketball hoop: The robot should be positioned close to the basketball hoop so that it can easily reach the ball when it falls through the hoop. 2. Wait for the ball to fall through the hoop: Once the ball falls through the hoop, the robot should wait for it to land on the ground. 3. Pick up the ball: The robot should pick up the ball as soon as it lands on the ground. 4. Dribble the ball towards the hoop: The robot should dribble the ball towards the hoop, aiming for the basket. 5. Shoot the ball into the hoop: The robot should shoot the ball into the hoop, attempting to dunk it. 6. Celebrate the successful dunk: If the robot successfully dunks the ball, it should celebrate its achievement.

**LLaMA-Adapter V2:** The image shows a basketball hoop with a basketball inside it, and a robot is positioned near the hoop. The robot is likely programmed to dunk the basketball, so it needs to execute a series of steps to achieve this goal. Here is a possible plan for the robot to dunk the basketball: 1. The robot should first identify the basketball and determine its position relative to the hoop. 2. The robot should then move towards the basketball, adjusting its speed and direction to ensure it reaches the ball at the right time. 3. As the robot approaches the basketball, it should extend its arm and grab the ball with its hand. 4. The robot should then lift its arm and place the basketball into the hoop, aiming for the basket. 5. Once the basketball is inside the hoop, the robot should release it.

**LLaVA:** The image shows a basketball court with a basketball hoop, a basketball, and a person holding a remote control. The scene appears to be a simulation or a video game, as the person is controlling the basketball with the remote. The basketball is positioned near the hoop, and the person is likely trying to dunk it. The remote control allows the person to simulate the dunking motion, which is not possible in real life.

**MiniGPT-4:** In this image, there is a basketball player standing on a wooden floor in front of a hoop. The player is holding a basketball in one hand and is preparing to dunk it through the hoop. The player is wearing a basketball uniform, and the hoop is hanging from the ceiling. There is a small orange ball lying on the ground next to the hoop, and a second ball is sitting on the top of the hoop. To write a plan for the robot to dunk the basketball, you could use the following steps: 1. Program the robot to move to the center of the room and stand in front of the hoop. 2. Program the robot to pick up the ball on the ground with its arm. 3. Program the robot to hold the ball with one hand while it uses its other hand to dunk the ball through the hoop. 4. Program the robot to release the ball after it has gone through the hoop. 5. Program the robot to move away from the hoop after it has finished dunking the ball. Overall, this plan would involve the robot using its arm to pick up the ball on the ground, its other hand to dunk the ball through the hoop, and then releasing the ball after it has gone through.

**mPLUG-Owl:** The image features a robot standing on a wooden floor and holding a basketball in its hand. It seems to be attempting to perform a dunk on a basketball hoop. To achieve this, the robot needs to first position itself properly and aim at the hoop. First, it should move towards the basketball hoop by walking on its wheels. It should maintain a straight path and avoid colliding with the walls or any obstacles in its way. Once it reaches the hoop, the robot should adjust its position and align its body with the rim. It should also maintain a stable posture while maintaining the basketball in its hand. Next, the robot should jump up and extend its arm and hand towards the hoop, while simultaneously bending its knees. This will generate enough momentum to launch the basketball toward the rim. As the basketball is in its grasp, the robot should release it with a quick flick of its wrist. The ball's trajectory should follow the path of the robot's arm, and it should enter the basket with a satisfying dunk.

**Otter:** The image shows a small toy basketball on a basketball court, with a large toy head of a man, which appears to be a statue or a decoration, nearby. The scene suggests that the toy robot is designed to perform a dunk on the basketball using a ramp. To accomplish this, the robot would need to follow a plan that involves moving to the appropriate position on the court, using the ramp to gain elevation, and timing its jump to make contact with the basketball at the right moment. The plan would likely involve adjusting the robot's speed, angle, and trajectory to achieve a successful dunk while maintaining balance and coordination throughout the process. The specific details of the plan would vary depending on the design and programming of the robot, but the general idea would be to mimic the motion of a human dunking the ball.

**VPGTrans:** The image shows a basketball hoop in the middle of a basketball court. The hoop is made of metal and has a net attached to it. There is a basketball on the ground next to the hoop. The hoop is at a height of about 10 feet off the ground. The basketball is at a height of about 5 feet off the ground. There is a robot on the left side of the hoop. The robot is facing the hoop and has a net attached to its arm. The robot is at a height of about 3 feet off the ground. The robot's arm is extended and ready to shoot the basketball into the hoop.

# D    Summary of Evaluation Datasets

 We summarize the evaluation datasets in our LVLM-eHub in Table A.6.

Table A.6: Description of datasets used in our LVLM-eHub.

| Dataset Name | Dataset Description | Evaluation Data |
|---|---|---|
| | Visual Perception Datasets | |
| ImageNet1K | The ImageNet1K dataset [54] consists of 1K object classes and contains 1,281,167 training images, 50 images per class for validation, and 100 images per class for testing. | 50K (val.) |
| CIFAR10 | The CIFAR10 dataset [26] has 10 classes and 6000 images per class with 5000 for training and 1000 for testing. | 10K (test) |
| Pets37 | The Oxford-IIIT Pet dataset [27] comprises 37 categories (*Pets37* for short) with 25 dog breeds and 12 cat ones and 200 images per class. There are 7349 images in total, 3680 trainval images, and 3669 test images. | 3669 (test) |
| Flowers102 | The Oxford 102 Flower dataset [28] includes 120 flower categories (*Flowers102* for short) with 40 to 258 images for each class and 8189 images in total, namely 10 images per class for both train and val and the rest for a test. | 6149 (test) |
| COCO-OC | We ask the model to count the number of a certain object appearing in the image and attend to individual objects, which is decoupled from high-level semantics and thus a more appropriate test bed for fine-grained visual understanding evaluation. We construct the dataset of this problem with images from the validation set of MSCOCO | 10000 (val) |
| COCO-MCI | We ask the model if a certain object exists in the image and attend to individual objects, which is decoupled from high-level semantics and thus a more appropriate test bed for fine-grained visual understanding evaluation. We construct the dataset of this problem with images from the validation set of MSCOCO | 10000 (val) |
| VCR-OC | Same as COCO-OC, but using images from the validation set of the VCR dataset | 10000 (val) |
| VCR-MCI | Same as COCO-MCI, but using images from the validation set of the VCR dataset | 10000 (val) |
| | Visual Knowledge Acquisition Datasets | |
| IIIT5K | The IIIT5K [30] is an ocr dataset that contains words from street scenes and originally-digital images. It is split into 2k/3k for train/test set. | 3k (test) |
| IC13 | The ICDAR 2013 dataset [31] consists of 229 training images and 233 testing images, with word-level annotations provided. Specifically, it contains 848 and 1095 cropped text instance images for the train and test sets respectively. | 848 (train) |
| IC15 | The ICDAR 2015 dataset [32] contains 1500 images: 1000 for training and 500 for testing. Its train/test set contains 4468/2077 cropped text instance images. | 2077 (test) |
| Total-Text | The total-test dataset [33] contains 1555 images: 1255 for training and 300 for testing. It contains 2551 cropped text instance images in the test set. | 2551 (test) |
| CUTE80 | The CUTE80 dataset [34] contains 288 cropped text instance images getting from 80 high-resolution images. | 288 (all) |
| SVT | The Street View Text (SVT) dataset [35] was harvested from google street view. It contains 350 images in total and 647 cropped text instance images for testing. | 647 (test) |
| SVTP | The SVTP dataset [36] contains 645 cropped text instance images. It is specifically designed to evaluate perspective-distorted text recognition. No train/test split was provided. | 645 (all) |
| COCO-Text | The COCO-Text dataset [37] we use is based on the v1.4 annotations, which contains 9896/42618 annotated words in val/train set. | 9896 (val) |
| WordArt | The WordArt dataset [38] consists of 6316 artistic text images with 4805 training images and 1511 testing images. | 1511 (test) |
| CTW | The SUCT-CTW1500 (CTW) dataset includes over 10,000 text annotations in 1500 images (1000 for training and 500 for testing) used in curved text detection. In our evaluation, we use 1572 rectangle-cropped images getting from the testing set. | 1572 (test) |
| HOST | The heavily occluded scene text (HOST) in Occlusion Scene Text (OST) dataset [40]. | 2416 (HOST) |
| WOST | The weakly occluded scene text (WOST) in the OST dataset. | 2416 (WOST) |
| SROIE | The SROIE dataset [41] contains 1000 complete scanned receipt images for OCR and KIE tasks. The dataset is split into 600/400 for the trainval/test set. In the KIE task, it is required to extract company, data, address, and total expenditure information from the receipt and there are 347 annotated receipts in the test set. | 347 (test) |
| FUNSD | The FUNSD dataset [42] contains 199 real, fully annotated, scanned forms for the KIE task. It is split 50/149 for the test/train set. | 50 (test) |

| Dataset Name | Dataset Description | Evaluation Data |
|---|---|---|
| NoCaps | The NoCaps dataset contains 15100 images with 166100 human-written captions for novel object image captioning. | 4500 (val) |
| Flickr-30k | The Flickr30k dataset consists of 31K images collected from Flickr, each image has five ground truth captions. We use the test split which contains 1K images. | 1K (test) |
| | Visual Reasoning Datasets | |
| DocVQA | DocVQA [65] contains 12K images and 50K manually annotated questions and answers. | 5349 (val) |
| TextVQA | Notably, we use the latest v0.5.1 version of TextVQA [80] dataset. It contains 34602 questions based on 21953 images from OpenImages' training set. Its validation set contains 5000 questions based on 3166 images. | 5000 (val) |
| STVQA | Scene Text Visual Question Answering (STVQA) [81] consists of 31,000+ questions across 23,000+ images collected from various public datasets. It contains 26074 questions in the train set and we sample 4000 samples from the train set in default order with seed 0. | 4000 (train) |
| OCR-VQA | OCRVQA [82] contains 100037 question-answer pairs spanning 207572 book cover images. | 100037 (all) |
| OKVQA | OKVQA [83] is a dataset about outside knowledge visual question answering. It contains 14055 open-ended question-answer pairs in total. | 5046 (val) |
| GQA | GQA [84] is a visual question-answering dataset with real images from the Visual Genome dataset. | 12578 (testdev) |
| Visdial | Visual Dialog (Visdial) [87] contain images sampled from COCO2014 and each dialog has 10 rounds. In our evaluation, we treat it as a VQA dataset by splitting each dialog sample into question-answer pairs by rounds. As there are 2064 dialog samples in the validation set, we have 20640 question-answer pairs collected from the validation set. | 20640 (val) |
| IconQA | IconQA dataset [69] provide diverse visual question-answering samples and we use the test set in its multi-text-choice task. | 6316 (test) |
| VSR | Visual Spatial Reasoning (VSR) dataset [70] contains a collection of caption-image pairs with true/false labels. We treat it as a VQA dataset by asking the model to answer True or False. | 10972 (all) |
| ScienceQA IMG | ScienceQA [46] is a multimodal benchmark containing multiple choice questions with a diverse set of science topics. In our evaluation, we only use the samples with images in the test set. | 2017 (test) |
| VizWiz | VizWiz [47] is a VQA dataset whose answers are got by asking blind people. | 1131 (val) |
| SNLI-VE | SNLI-VE[45] extends the text entailment (TE) task into the visual domain and asks the model whether the image is semantically entailed, neutral, or contradicted to the next hypothesis. It is a three-category classification task based on Flicker30k[88]. | 500 (val) |
| | Visual Commonsense Datasets | |
| ImageNetVC | ImageNetVC[15] is a fine-grained human-annotated dataset for zero-shot visual commonsense evaluation, containing high-quality QA pairs across diverse domains with sufficient image sources. | 10000 (rank) |
| VCR | VCR [48] is a challenging multiple-choice VQA dataset that needs commonsense knowledge to understand the visual scenes and requires multiple-steps reasoning to answer the question. | 500 (val) |
| | Object Hallucination Datasets | |
| COCO-Random | Following [14], we randomly select 500 images from the validation set of MSCOCO with more than three ground-truth objects in the annotations and construct 6 questions for each image. The probing objects in the questions that do not exist in the image are randomly sampled | 3000(val) |
| MSCOCO-Popular | Similar to COCO-Random, we randomly select 500 images and construct 6 questions for each image. But the probing objects in the questions that do not exist in the image are selected from the top-50% most frequent objects in MSCOCO [14]. | 3000(val) |
| MSCOCO-Adversarial | Similar to COCO-Random, we randomly select 500 images and construct 6 questions for each image. But the probing objects in the questions that do not exist in the image are selected from the ranked objects with their co-occurring frequency and the top-50% most frequent objects are sampled [14]. | 3000(val) |
| | Embodied Intelligence Datasets | |
| Embodied AI Tasks | Minecraft [50], VirtualHome [51], Meta-World [52], and Franka Kitchen [52] | selected samples |