# OpenReview forum: "LVLM-eHub: A Comprehensive Evaluation Benchmark for Large Vision-Language Models"
_NeurIPS.cc/2023/Track/Datasets_and_Benchmarks — Submitted to NeurIPS 2023 Datasets and Benchmarks_

### Official Review · Reviewer_Z9KL · 2023-06-25

**Rating:** 6
**Confidence:** 3

**Strengths:**

1. The evaluation problem paper aims to tackle is an important and timely one. There are newer LVLMs being released constantly, and it is important to have a benchmark that would help the community understand the performance/tradeoffs with using a model.

2. The evaluation set and the model set has a large breadth. There are many multimodal tasks in the evaluation set, along with very recent models in the model set. The released repository presents a good opportunity for the community

3. The codebase is accessible, and I believe it would be useful to the community.

**Additional Feedback:**

1. Could the authors elaborate a bit more about how the chat evaluation works? The description in L130-133 reads a bit confusing, for instance, what are “single round chats”?

2.  In L154, what do the authors mean by “The likelihood of image-text pair can be generated”?

3.  In Appendix L644, it says “see Appendix”

4. Am I correct to understand that the mPLUG-OWL performance in Table 6 is below 50%, which is the random chance to the Yes/No question? If the task is not balanced, what is the chance-level performance?

5. The repository says “ Then for each model, they may require conflicting versions of python packages, we recommend creating a specific environment for each model based on their GitHub repo.”. It would be really great to have the repo self-contained with the list of requirements for models.

[Grammar] Abstract, “generate objects” -> “generated objects”
[Grammar] L183:  “making it overfitting on these tasks” -> “making it overfit to these tasks”.

**Clarity:**

I believe the clarity of the writing could be improved on two fronts. For one, there are parts of the paper that would be better off with a mathematical description, e.g. the prefix-based score (L152) description is vague to me. Secondly, I am unclear whether some of the claims are hypotheses or sufficiently justified statements, such as the ones in my comments 8-9-11, I would have appreciated either posing them as hypotheses or providing less ambiguous justification.

**Correctness:**

Some of the statements lack sufficient justification in my opinion. I tried to highlight them in my comments above in the “Opportunities for Improvements” Section, e.g. Comments 8-9-11. Similarly, I think the paper needs a better description of the metrics used (e.g. what is executability, what is the "Popular" dataset in Table 7) and the concrete generation procedures with the model (e.g. what is the temperature? do all models have the same sampling methods? what are hyperparameters?).

**Documentation:**

The repository and datasets/models are released and accessible. I believe documentation is reasonable, yet could be improved (e.g. the entry points to models, model APIs, dataset APIs).

**Ethics:**

Apologies if I missed this, but I am not clear whether the user studies have the necessary Institutional Review Board approvals (see https://neurips.cc/public/EthicsGuidelines under "Research involving human participants"). If the IRB Approval is not necessary for an approved reason or the necessary approval is taken from the IRB, this should be highlighted in the manuscript.

**Limitations:**

I do not believe the limitations of the evaluation methodology is well-discussed. For instance, see my comment above “Opportunities for Improvements” Section comments 3-4.

**Opportunities For Improvement:**

1. Compositionality is slightly overlooked here. Quite a few recent works (e.g. [1, 2, 3]) demonstrate how models could struggle with compositional reasoning/exhibit bag-of-words-like behavior. It would be good to see a few of these benchmarks, e.g. ARO[1] or VL-Checklist[2].

2. I believe formalizing the tasks and setting up mathematical notation could help improve the precision of the communication. For instance, I do not exactly understand how a prefix-based score (L152) actually works, and having a mathematical description would be very helpful. A similar suggestion goes for multi-turn reasoning.

3. I am unclear whether some of these evaluations are fair enough. For instance, in image classification, authors use a single prompt “The photo of the”, and evaluate all models with this prompt. The sensitivity of LMs or VLMs to the prompts is very well-known, e.g. see the recent discussion in [5]. The common way one performs zero-shot classification is at the very least to try multiple prompts[4] and one expects performance degradation otherwise. This is, in my view, a significant limitation that either needs addressing or discussing in the main text.

4. In addition to the above, some of the performance metrics are error-prone to me. E.g. for image classification, it is based on a single sample from the model, see “we considered the prediction as correct if the model output contains the correct class name”(L647). Even though it is based on a single sample, the authors do not describe the sampling procedure from the model (is the temperature 0? Otherwise, is a single sample reliable?). Similarly, some of the metrics are not defined (e.g. such as, what is “Executability” in Table 7?) It would improve the presentation to concretely describe the sampling procedures and metrics.

5. In my opinion, it is a bit hard to extract actionable insights from the paper. To be more concrete, from such a large-scale benchmark, personally, I would expect reading through the insights such as “Models X,Y are better in Domain D”; “The methodology M significantly hurts the performance in setting S”. These pieces of information do exist in the manuscript in a scattered way, but I believe it would benefit the presentation to have these recommendations more localized and concise somewhere like the end of the introduction.

6. Apologies if I miss this, but does the user study have an IRB Approval? Who are the users, how was the user study conducted, is there a description of the interface? I believe the paper would benefit from a better description of the user studies that are conducted.

### Minor
7. L180: What is “coarse-fined classification”?

8. There are a few strong claims that are not backed up enough, in my opinion. E.g. L182: The comment “The main reason is that InstructBLIP is fine-tuned on 1.6M VQA data” is a strong claim, I would recommend posing it as a hypothesis. Similar one in L196 “The main reason is that InstructBLIP overfits these tasks by fine-tuning massive VQA data.”

9. L247: “Based on the findings, two deductions can be made. Firstly, the use of image-text pairs is consequential in aligning visual-text features. This is evident from the comparison demonstrated in Table 1.” How is this evident from Table 1?

10. Table 6: What is random/popular/adversarial? This is not mentioned neither in the main text nor in the Appendix.

11. L235: “Such object hallucination problem can be alleviated by a multi-turn reasoning pipeline shown in the experiments on SNLI-VE and VCR.” How can we infer this from the results in the paper?

12. Section header for Section 4 says “Discussion and Conlcusion”

13. There are many important details in the Appendix such as the concrete evaluation metrics, which I believe the readers would benefit from having them in the main text.

[1] Yuksekgonul, Mert, et al. "When and why vision-language models behave like bags-of-words, and what to do about it?." arXiv e-prints (2022): arXiv-2210.

[2] Zhao, Tiancheng, et al. "VL-CheckList: Evaluating Pre-trained Vision-Language Models with Objects, Attributes and Relations." arXiv preprint arXiv:2207.00221 (2022).

[3] Parcalabescu, Letitia, et al. "VALSE: A task-independent benchmark for vision and language models centered on linguistic phenomena." arXiv preprint arXiv:2112.07566 (2021).

[4] https://github.com/openai/CLIP/blob/main/notebooks/Prompt_Engineering_for_ImageNet.ipynb

[5] https://huggingface.co/blog/evaluating-mmlu-leaderboard

**Relation To Prior Work:**

I believe related works are comprehensive and well-discussed.

**Summary And Contributions:**

LVLMs are advancing fast, and it is hard to understand the tradeoffs between the models as evaluation sets often differ. This work proposes a comprehensive evaluation suite to better understand LVLM performance. The suite includes a large variety of tasks and recent models, ranging from image classification and captioning to commonsense reasoning, embodied learning, and beyond. The released repository provides a valuable resource to the community.

---

> ### Author Response · Authors · 2023-08-15
> **Response to Reviewer Z9KL Part(1/5)**
>
> Thank you for your valuable comments and insightful suggestions. We have carefully considered each of your queries and addressed them as follows:
>
> **Q1.** [Compositionality is slightly overlooked here. Quite a few recent works (e.g. [1, 2, 3]) demonstrate how models could struggle with compositional reasoning/exhibit bag-of-words-like behaviour. It would be good to see a few of these benchmarks, e.g. ARO[1] or VL-Checklist[2].]
>
> **A1.** Thanks to the reviewer for the suggestion. We evaluated the performance of eight models in the ARO benchmark, which includes VG_Relation, VG_Attribution, COCO_Order, and Flickr30k_Order. In each subset of the ARO benchmark, we randomly selected 100 samples with seed 0. Different from VLMs such as BLIP and CLIP, which predict the answer by comparing the similarities between the image feature and text features, LVLM obtains the final answer by transferring the ARO benchmark into a multi-choice visual question answering task. For example, given an image showing that the horse is eating the grass, we give the LVLM model two choices: A) the horse is eating the grass, and B) the grass is eating the horse. We report the final accuracy in Table A. We can see that existing LVLMs exhibit intriguing deficiencies in understanding compositionality. Only Instruct BLIP, LLaMA-Adapter V2, and MiniGPT-4 exceed the chance-level accuracy on all benchmarks.  There are even no LVLMs outperforming VLM model X-VLM in terms of the average score on all ARO benchmarks. Hence, it has plenty of room to improve the comprehension of LVLMs in compositionality such as relation, attribute, and order.
>
> **Table A.** The results of LVLMs on the ARO benchmark, which assesses the understanding of relation, attribution, and order.
> |              | VG_Relation | VG_Attribution | COCO_Order | Flickr30k_Order | Avg. |
> | ------------ | ----------- | -------------- | ---------- | --------------- | ---- |
> | Chance Level | 50          | 50             | 20         | 20              | -    |
> | VLM Baseline |             |                |            |                 |      |
> | CLIP         | 51          | 55             | 50         | 62              | 54.5 |
> | BLIP         | 57          | 83             | 37         | 44              | 55.3 |
> | X-VLM        | 65          | 89             | 35         | 56              | 61.3 |
> | LVLM         |             |                |            |                 |      |
> | BLIP2        | 45          | 96             | 21         | 45              | 51.8 |
> | InstructBLIP | 73          | 83             | 31         | 44              | 57.8 |
> | LA-V2        | 60          | 70             | 47         | 55              | 58.0 |
> | LLaVA        | 42          | 47             | 60         | 31              | 45.0 |
> | MiniGPT-4    | 63          | 63             | 42         | 27              | 48.8 |
> | mPLUG-Owl    | 37          | 64             | 11         | 9               | 30.3 |
> | Otter        | 5           | 45             | 12         | 16              | 19.5 |
> | VPGTrans     | 47          | 69             | 33         | 20              | 42.3 |
>
> **Q2.** [I believe formalizing the tasks and setting up mathematical notation could help improve the precision of the communication. For instance, I do not exactly understand how a prefix-based score (L152) actually works, and having a mathematical description would be very helpful. A similar suggestion goes for multi-turn reasoning.]
>
> **A2.** Thanks to the reviewer for the suggestion.  We briefly introduce the prefix-based score and multi-turn reasoning here. Please also see Sec.A.3 of the supplementary revision.
>
> The prefix-based score is the probability of generating the candidate's answer. It feeds the same image together with each option of a multi-choice visual question respectively and independently, and gathers the logarithm of probabilities of the candidate option, resulting in a holistic loglikelihood of that option given the image.
>
> In addition,  multi-turn reasoning consists of three components in the framework. Firstly, a Questioner (CharGPT) decomposes the main question into several sub-questions, then an Answerer (LVLM) answer all sub-questions with detailed analysis, and finally, a Reasoner (ChatGPT) analyzes both sub-questions and sub-answers to decide if a confident answer to the main question can be derived. Such a procedure works iteratively.

---

> > ### Author Response · Authors · 2023-08-15
> > **Response to Reviewer Z9KL Part(2/5)**
> >
> > **Q3.** [I am unclear whether some of these evaluations are fair enough. For instance, in image classification, authors use a single prompt “The photo of the”, and evaluate all models with this prompt. The sensitivity of LMs or VLMs to the prompts is very well-known, e.g. see the recent discussion in [5]. The common way one performs zero-shot classification is at the very least to try multiple prompts[4] and one expects performance degradation otherwise. This is, in my view, a significant limitation that either needs addressing or discussing in the main text.]
> >
> > **A3.** Thank you for your insightful question. It is widely acknowledged that both Language Models (LMs) and Vision-and-Language Models (VLMs) demonstrate sensitivity to prompts. We have pointed it out in Sec. C.4 of the Appendix. To thoroughly explore the potential impact of the prompts we utilized on the performance of VLMs, we conducted an empirical study. This investigation aimed to provide insights into the performance of VLM models across a range of prompts. We employed a set of 80 prompts, previously utilized in the [prompt engineering experiment of CLIP](https://github.com/openai/CLIP/blob/main/notebooks/Prompt_Engineering_for_ImageNet.ipynb), and applied them to a subset of the ImageNet1K validation set. Within this subset, we randomly selected 3 images for each class, resulting in a total of 3,000 images. This comprehensive approach enabled us to analyze the nuanced performance variations of VLMs under diverse prompts. The detailed performance metrics are presented in Table B.  The term "baseline" refers to the accuracy of the prompt we used in our paper, while "mean" and "std" represent the statistical performance measures derived from the aforementioned set of 80 prompts. We can see that BLIP2 and InstructBLIP still achieve competitive performance on the ImageNet validation subset over 80 prompts.
> >
> > **Table B.** The results of LVLMs with 80 different prompts.
> > |                                | BLIP2     | InstructBLIP | LA-V2 | LLaVA | MiniGPT-4 | mPLUG-Owl | Otter | VPGTrans |
> > | ------------------------------ | --------- | ------------ | ----- | ----- | --------- | --------- | ----- | -------- |
> > | baseline (prompt in the paper) | *24.05*   | **24.4**     | 19.17 | 18.17 | 16.37     | 20.37     | 14.47 | 17.1     |
> > | mean (80 prompts)              | **26.52** | *24.30*      | 18.34 | 17.47 | 13.8      | 18.16     | 13.78 | 13.32    |
> > | std (80 prompts)               | 3.28      | 2.47         | 1.35  | 1.38  | 2.31      | 2.74      | 0.73  | 1.30    |
> >
> > **Q4.** [In my opinion, it is a bit hard to extract actionable insights from the paper. To be more concrete, from such a large-scale benchmark, personally, I would expect reading through the insights such as “Models X,Y are better in Domain D”; “The methodology M significantly hurts the performance in setting S”. These pieces of information do exist in the manuscript in a scattered way, but I believe it would benefit the presentation to have these recommendations more localized and concise somewhere like the end of the introduction.]
> >
> > **A4.** Thanks for the suggestion.  We can conclude some actionable insights from our evaluation results.  First, the quality of visual instruction data matters more than quantity in the open-world VQA. We observe that MiniGPT-4, which is tuned by only 3.5K high-quality visual instruction data performs much better than InstructBLIP tuned on visual instruction data adapted from various existing VQA datasets in our MultiModality Arena.  Second,  a strong visual encoder can help extract detailed information from the image, leading to good performance in OCR tasks. For instance, we see that BLIP2, InstructBLIP, and VPGTrans achieve better performance than the remaining 5 LVLMs. This may be because the visual encoder ViT-g/14 used in BLIP2, InstructBLIP, and VPGTrans is more powerful than ViT-L/14 employed in the remaining LVLMs. Third, multi-turn reasoning helps alleviate the hallucination issue, indicating that the evaluation method with critical thinking can induce the correct prediction from the model. We find that LVLM with multi-turn reasoning can determine whether an object exists in the image more accurately than single-turn reasoning. Hence, multi-turn reasoning is appropriate to assess the full potential of the model. Fourth, LVLMs tuned with high-quality instruction-following data present more promising planning ability than models without being tuned with instruction data as demonstrated in Table 7.
> >
> > We hope that these conclusions can further encourage more insights into developing advanced LVLMs. Since the existing models are usually different in network structure, training data and training recipes, we will investigate the effect of each component by rigorous ablation studies in the future version.

---

> > > ### Author Response · Authors · 2023-08-15
> > > **Response to Reviewer Z9KL Part(3/5)**
> > >
> > > **Q5.** [In addition to the above, some of the performance metrics are error-prone to me. E.g. for image classification, it is based on a single sample from the model, see “we considered the prediction as correct if the model output contains the correct class name”(L647). Even though it is based on a single sample, the authors do not describe the sampling procedure from the model (is the temperature 0? Otherwise, is a single sample reliable?). Similarly, some of the metrics are not defined (e.g. such as, what is “Executability” in Table 7?) It would improve the presentation to concretely describe the sampling procedures and metrics.]
> > >
> > > **A5.** Good question. The sampling parameter used in our quantitative evaluation is the same as it is in the LVLM arena, which also means that the temperatures are not zero for every model. Specifically, we use the default sampling parameter used in each model's GitHub repo, as such a parameter is usually a good choice for the underlying model tuned by the model provider.  To further test the sensitivity of LVLMs to randomness,  we run experiments on the ImageNet validation subset (i.e. 3 images for each class and 3k images in total)  three times with the same inference configuration. We can see that all LVLMs present a small accuracy variation as shown in Table C. This may be because the population accuracy on 3k images of ImageNet can converge despite the randomness of the single testing sample. Since there are usually thousands of image-text pairs of evaluation datasets in our LVLM-eHub as shown in Sec.D of the Appendix, we believe that our evaluation results are reliable.
> > >
> > > In addition, the executability of the planning represents the pragmatic feasibility of realizing the envisaged plan in tangible scenarios. In other words, the human annotator will determine whether the goal in the question can be achieved if the plan generated by LVLMs is followed. For example, consider the "Dunk the basketball" task from the Meta-World benchmark, as illustrated in Figure A.7 of the Appendix. The mPLUG-Owl model generated plans that exude reasonability, encompassing strategies such as navigating toward the hoop via wheels, extending the robotic arm, and subsequently releasing the ball. Contrarily, these delineated actions are non-executable for the depicted robotic arm, given its inherent fixed-base configuration.
> > >
> > > **Table C.** The sensitivity of the evaluation results of LVLMs to randomness. We run experiments on the ImageNet validation subset (i.e. 3 images for each class and 3k images in total)  ten times with the same inference configuration.
> > > |                  | BLIP2   | InstructBLIP | LA-V2 | LLaVA | MiniGPT-4 | mPLUG-Owl | Otter | VPGTrans |
> > > | ---------------- | ------- | ------------ | ----- | ----- | --------- | --------- | ----- | -------- |
> > > | Mean (10 rounds) | *23.87* | **24.40**    | 21.87 | 18.64 | 16.86     | 20.38     | 14.47 | 15.92    |
> > > | Std (10 rounds)  | 0.01    | 0.01         | 0.20  | 0.30  | 0.46      | 0.55      | 0.01  | 0.42     |
> > >
> > > **Q6.** [Apologies if I miss this, but does the user study have an IRB Approval? Who are the users, how was the user study conducted, and is there a description of the interface? I believe the paper would benefit from a better description of the user studies that are conducted.]
> > >
> > > **A6.** Thanks for the advice. Our user study asks users to evaluate the planning generated by different LVLMs.  The user study involves researchers and students in our research institute. Users are experienced in assessing the quality, relevance, and usefulness of the text generated by the LVLM in a specific context. The proficiency of participants guarantees reliable evaluation results. The user study questionnaire was conducted on the [WJX platform with Link](https://www.wjx.cn/vm/rmDbu01.aspx#).  On each page, the answers generated by 8 LVLMs are listed for the given visual question. The user is asked to give a score ranging from 1-5 for each model. For a fair comparison, we randomly shuffle the model’s output and anonymize outputs during evaluation. The evaluation results are available at our [GitHub repo](https://github.com/OpenGVLab/Multi-Modality-Arena/tree/main/LVLM_evaluation/utils_data/EmbodiedDataset). Note that our user study does not involve direct interactions with human participants and does not involve potential risks to participants, such as the collection of personal information or sensitive data, potential privacy concerns, or any other aspects that could impact the participants' rights or well-being. We do not necessarily have IRB Approval.
> > >
> > > **Q7.** [L180: What is “coarse-fined classification”?]
> > >
> > > **A7.** Sorry for any caused confusion. It's a typo and it should be "coarse-grained classification".

---

> > > > ### Author Response · Authors · 2023-08-15
> > > > **Response to Reviewer Z9KL Part(4/5)**
> > > >
> > > > **Q8.** [There are a few strong claims that are not backed up enough, in my opinion. E.g. L182: The comment “The main reason is that InstructBLIP is fine-tuned on 1.6M VQA data” is a strong claim, I would recommend posing it as a hypothesis. Similar one in L196 “The main reason is that InstructBLIP overfits these tasks by fine-tuning massive VQA data.”]
> > > >
> > > > **A8.** Thanks for the suggestion. We agree with the reviewer that these claims should be posed as a hypothesis. We state that InstructBLIP is fine-tuned on 16M VQA data because we count all instruction datasets in InstructBLIP （see Table 4 of [InstructBLIP Paper](https://arxiv.org/abs/2305.06500). However, it is unknown whether the entire samples of instruction datasets are used. Hence, it is better to present them with a hypothesis, as suggested by the reviewer. We will correct it in the final version.
> > > >
> > > > **Q9.** [L247: “Based on the findings, two deductions can be made. Firstly, the use of image-text pairs is consequential in aligning visual-text features. This is evident from the comparison demonstrated in Table 1.” How is this evident from Table 1?]
> > > >
> > > > **A9.** We are sorry for the wrong deduction.  From Table 1,  we observe that Otter and InstructBLIP are not pre-trained in paired image-text data. Then we make a deduction that image-text pairs are important. However, we ignore that these two models load pre-trained LVLMs such as BLIP2 and Flamingo which have been trained in massive image-text pairs. Thanks for pointing out this. We will remove the wrong deduction in the final version.
> > > >
> > > > **Q10.** [Table 6: What is random/popular/adversarial? This is not mentioned neither in the main text nor in the Appendix.]
> > > >
> > > > **A10.** We are sorry for the confusion.  We have provided the descriptions of the datasets we used in Appendix D.  Following POPE [1], we use COCO Random/Popular/Adversarial to investigate the hallucination of LVLMs.  We randomly select 500 images with more than three ground-truth objects in the annotations. Each image is paired with 6 Yes/No questions where Yes/No indicates whether an object exists (Yes) in the image or not (No). For VQA samples with the answer of No, we create them in three ways. (1) COCO Random randomly samples the objects that do not exist in the image. (2) COCO Popular selects the top-k most frequent objects in MSCOCO that do not exist in the image. And (3) COCO Adversarial first ranks all the objects according to their co-occurring frequencies with the ground-truth objects and then selects the top-k most frequent ones that do not exist in the image.
> > > > [1] Yifan Li et al. Evaluating Object Hallucination in Large Vision-Language Models.
> > > >
> > > > **Q11.** [L235: “Such object hallucination problem can be alleviated by a multi-turn reasoning pipeline shown in the experiments on SNLI-VE and VCR.” How can we infer this from the results in the paper?]
> > > >
> > > > **A11.** We are sorry for the insufficient explanations. From Table 6,  we know that LVLMs tuned by visual instruction data suffer from object hallucination issues because they achieve lower accuracy than BLIP2 on COCO Random/Popular/Adversarial using single-turn reasoning evaluation (one-round VQA).  Moreover,  we find that LVLMs tuned by visual instruction data consistently outperform BLIP2 on both VCR and SNLI-VE using multi-turn reasoning evaluation (see Table 4 and Table 5).  Hence, we infer that the object hallucination problem can be alleviated by a multi-turn reasoning pipeline shown in the experiments on SNLI-VE and VCR. The inference is indirect as the results are obtained from separate benchmarks.
> > > >
> > > > To further verify this, we directly evaluate the performance of LVLMs on COCO Random using multi-turn reasoning. For quick verification, we sample 50 samples from COCO Random. The results are reported in Table  D. We can see that the performances of LVLMs improve a lot under multi-turn reasoning, indicating that multi-turn reasoning can alleviate the issue of object hallucination.
> > > >
> > > > **Table D.** The performance of LVLMs on COCO Random evaluated by multi-turn reasoning. Multi-turn reasoning can alleviate the issue of object hallucination.
> > > > |             | BLIP2 | InstructBLIP | LA-V2 | LLaVA | MiniGPT-4 | mPLUG-Owl | Otter |
> > > > | ----------- | ----- | ------------ | ----- | ----- | --------- | --------- | ----- |
> > > > | Single-turn | 72    | 80           | 72    | 58    | 56        | 48        | 60    |
> > > > | Multi-turn  | 74    | 80           | 76    | 74    | 66        | 80        | 68    |

---

> > > > > ### Author Response · Authors · 2023-08-15
> > > > > **Response to Reviewer Z9KL Part(5/5)**
> > > > >
> > > > > **Q12.** [Section header for Section 4 says “Discussion and Conlcusion”]
> > > > >
> > > > > **A12.** Sorry for the confusion, we will change it to "Discussion and Conclusion" in our revised version.
> > > > >
> > > > > **Q13.** [There are many important details in the Appendix such as the concrete evaluation metrics, which I believe the readers would benefit from having them in the main text.]
> > > > >
> > > > > **A13.** Thank you for your suggestion. We place these details in the Appendix as the main pages are limited to nine. We will add a section to present the concrete evaluation metrics for clarity.
> > > > >
> > > > > **Q14.** [Could the authors elaborate a bit more about how the chat evaluation works? The description in L130-133 reads a bit confusing, for instance, what are “single round chats”?]
> > > > >
> > > > > **A14.** We are sorry for the insufficient descriptions. The online chat evaluation contains three primary steps, which are matchmaking, single-round chat, and user voting. In the matchmaking step, we select two models from all eight models evaluated in our paper at random. Then in the single-round chat step, sampled models are asked to answer a question with a given visual input. After two sampled models generate their answers, users are requested to vote for their preferred model. As the sampled models in our online chat evaluation are only requested to answer once, we also name it a single round chat.
> > > > >
> > > > > **Q15.** [In L154, what do the authors mean by “The likelihood of image-text pair can be generated”?]
> > > > >
> > > > > **A15.** We are sorry for the vague statement.  Please see Q2/A2 for more details.
> > > > >
> > > > > **Q16.** [In Appendix L644, it says “see Appendix”]
> > > > >
> > > > > **A16.** Sorry for the confusion. We will delete it in our revised version.
> > > > >
> > > > > **Q17.** [Am I correct to understand that the mPLUG-Owl performance in Table 6 is below 50%, which is the random chance to the Yes/No question? If the task is not balanced, what is the chance-level performance?]
> > > > >
> > > > > **A17.** Good question.  The chance-level  performance should be 50%. It is true that mPLUG-Owl performance shown in Table 6 is below 50%. Firstly, we check the Yes/No ratio of gt answers for the MSCOCO Random/Popular/Adversarial datasets, and their Yes/No ratio is 1.06, 1.0, and 1.0 respectively.  Hence, the samples in the dataset are balanced. Then, after calculating the ratio of answering yes and no in the responses of mPLUG-Owl, we found that mPLUG-Owl is more inclined to answer yes. Therefore, we thought that maybe the mPLUG-Owl model has a performance below 50% because it is prone to be hallucinated, i.e. answer no even if the object exists in the image and answer yes even if the object does not exist in the image.
> > > > >
> > > > > **Q18.** [The repository says “ Then for each model, they may require conflicting versions of python packages, we recommend creating a specific environment for each model based on their GitHub repo.”. It would be really great to have the repo self-contained with the list of requirements for models.]
> > > > >
> > > > > **A18.** Good question. We have reorganized the original code of the model we used and prepared the list of requirements which can create a Python environment that works for all models used in our paper. The detailed information is shown in https://github.com/OpenGVLab/Multi-Modality-Arena/tree/main/LVLM_evaluation

---

> > > > > > ### Comment · Reviewer_Z9KL · 2023-08-21
> > > > > > **Thank you for your rebuttal**
> > > > > >
> > > > > > I thank the authors for a detailed rebuttal and experiments to respond to our questions in detail. A good fraction of my concerns are resolved. Below, I have a few more clarification questions.
> > > > > >
> > > > > > > Results in the Rebuttal
> > > > > >
> > > > > > Do the authors consider adding the results in the rebuttal to the paper? If I’m not missing it, I do not see some of these in the main paper / supplementary material. I do believe these justify some of the design choices of the benchmark (e.g. A3) or as I mentioned the important facets of evaluation (A1). I believe the inclusion of this content would benefit the paper, it would be good to see it in the revision.
> > > > > >
> > > > > > > Extracted insights
> > > > > >
> > > > > > I appreciate the findings that you convey in A4. Personally, I do believe it would benefit the reader to easily access these interesting findings in the paper, e.g. in a localized way at the end of the intro section / the discussion section.
> > > > > >
> > > > > > >  A5 - Setting of sampling ablation
> > > > > >
> > > > > > I think the randomness would be more interesting to observe in longer-form generation tasks, for 0-shot classification the generations are pretty short if I’m not wrong? Otherwise I’m not sure what to infer from the results in A5
> > > > > >
> > > > > > > A6 - User Study
> > > > > >
> > > > > > I do believe these details should be in the paper (if not there already, I could not find them). Quoting from the Ethics Review guidelines: `if the research presented involves direct interactions between the researchers and human participants or between a technical system and human participants, authors are required to follow existing protocols in their institutions (e.g. human subject research accreditation, IRB) and go through the relevant process.`
> > > > > >
> > > > > > If the study involves human participants, the authors are required to have an institutional approval and it is not up to the judgment of the scientists performing the study, per my understanding. I will defer this point to the AC’s judgment.
> > > > > >
> > > > > > > Overall Comment
> > > > > >
> > > > > > I am slightly leaning towards an acceptance position after the authors’ detailed rebuttal. Other than the above questions, I would be interested to see these suggested changes reflected in the submission to make sure that the important details are included in the submission.

---

> > > > > > > ### Author Response · Authors · 2023-08-28
> > > > > > > **Thanks for your response**
> > > > > > >
> > > > > > > Thank the reviewer for the kind reply and valuable comments.  We appreciate your engagement with our rebuttal and the points you've raised.  We address the remaining concerns as follows.
> > > > > > >
> > > > > > > > The results in the Rebuttal
> > > > > > >
> > > > > > > We appreciate your suggestion about our manuscript. We have integrated these components into the revised version. Specifically,
> > > > > > > 1. The experiments about compositionality are added in  Sec. C.3 of Appendix.
> > > > > > > 2. The formulations of the prefix-based score and multi-turn reasoning are given in Sec. A.3 of Appendix.
> > > > > > > 3. The experiments about the sensitivity of LVLMs to the prompts are presented in Sec. C.1 of Appendix.
> > > > > > > 4. The experiments about the randomness of LVLMs are presented in Sec. C.3 of Appendix.
> > > > > > > 5. The takeaway notes are provided in Sec.3.8.
> > > > > > > 6. More discussions about the ethical concerns of our user study are added in Lines 170-174 in Sec. 2.3.
> > > > > > > 7. Typos in the original manuscript are corrected.
> > > > > > > 8. More experiments that can show that multi-turn reasoning can alleviate the issue of hallucination are included in Sec. C.2 in Appendix.
> > > > > > > 9. Full code and evaluation pipeline is released in our open-source Github repo.
> > > > > > >
> > > > > > > We will further carefully check each advice given by the reviewer. Please also let us know if we missed any pieces of your interest.
> > > > > > >
> > > > > > > > Extracted insights
> > > > > > >
> > > > > > > We're glad to hear that you found the insights in section A4 valuable. Your suggestion could indeed provide readers with a clear overview of our significant findings. We have carefully integrated these insights in Sec.3.8.
> > > > > > >
> > > > > > > > A5 - Setting of sampling ablation
> > > > > > >
> > > > > > > Thanks for the advice. To elaborate, our evaluations further encompass the image captioning task on NoCaps dataset which consists of 4.5K samples. The image captioning task notably generates long-form responses. The detailed performance metrics are comprehensively presented in the subsequent table. In Table E, the term "baseline" corresponds to the results outlined in our paper, while "mean" and "std" indicate the statistical performance averaged over three repetitions.  We can see that all LVLMs present a small score variation from Table E.
> > > > > > >
> > > > > > > **Table E.** Evaluation of NoCaps dataset. Performance measured in CIDEr score.
> > > > > > > |                                | BLIP2     | InstructBLIP | LA-V2 | LLaVA | MiniGPT-4 | mPLUG-Owl | Otter | VPGTrans |
> > > > > > > | ------------------------------ | --------- | ------------ | ----- | ----- | --------- | --------- | ----- | -------- |
> > > > > > > | baseline (reimplementation)           | 48.6   | 46.6   | 33.7 | 1.6 | 5.8     | 0.3     | 11.6 | 36.2     |
> > > > > > > | mean (3 rounds)           | 48.8 | 46.2      | 33.8 | 1.2 | 6.4      | 0.2     | 11.7 | 36.6    |
> > > > > > > | std (3 rounds)               | 0.01      | 0.01         | 0.14  | 0.11  | 1.13      | 0.04      |0.01  | 0.56    |
> > > > > > >
> > > > > > > > A6 - User Study
> > > > > > >
> > > > > > > Thanks for the advice. More discussions about the ethical concerns of our user study are added in Lines 170-174 in Sec. 2.3. We are dedicated to addressing the ethical and moral considerations regarding the user evaluation method with thoroughness and commitment, while also providing effective solutions.
> > > > > > >
> > > > > > > > Overall Comment
> > > > > > >
> > > > > > > We're pleased to hear that our detailed rebuttal has positively influenced your perspective on our work. Your leaning towards an acceptance position is encouraging, and we're committed to addressing the points you've raised to enhance the quality and completeness of our final version.

---

### Official Review · Reviewer_r23C · 2023-07-02

**Rating:** 5
**Confidence:** 4
**Correctness:** The evaluation metrics seem appropriate.
**Clarity:** The paper is well-written and easy to…

**Strengths:**

(1) The paper proposed an evaluation benchmark of 8 representative LVLMs.

(2) The quantitative capability evaluation aspects are representative as well.

(3) The paper also included findings based on the benchmark.


**Additional Feedback:**

Please see the comments above.

**Documentation:**

Code is provided in a public GitHub repo.



**Limitations:**

Based on the large number of tasks the authors focused, it may be unclear whether the selected 8 models are sufficient enough to support the claim of the findings.


**Opportunities For Improvement:**

(1) It’s a little bit unclear how the models are selected, and why they are representative enough. It would be good to add an explanation about that.

(2) Each model was pretrained by different data, VE, LLM, and adaptor, it may be good to make a roughly fair comparison to provide some findings about the influence of each component.

(3) It would be good to have a conclusion, if this benchmark can provide some inspiration about which architecture, which training mechanism, etc, can lead to better performance.

(4) One important question could be, after building the benchmark, could the authors provide any potential ways that can improve the performance further?


**Relation To Prior Work:**

The literature review seems comprehensive.


**Summary And Contributions:**

This paper presents an evaluation of publicly available large multimodal models of their efficacy, which include 8 representative LVLMs. By building the benchmark, the paper provided findings from the evaluation results as well.

---

> ### Author Response · Authors · 2023-08-15
> **Response to Reviewer r23C Part (1/2)**
>
> Thank you for your valuable comments and insightful suggestions. We have carefully considered each of your queries and addressed them as follows.
>
> **Q1.** [It’s a little bit unclear how the models are selected, and why they are representative enough. It would be good to add an explanation about that.]
>
> **A1.** Thank you for the suggestion. it's important to note that the selected 8 LVLMs are the most up-to-date, open-source, and re-implementable at the commencement of our work, specifically around the time of our onboarding onto the online arena on May 13th.  In addition, LVLM is typically constructed by incorporating a Large Language Model (LLM) with a pre-trained visual encoder which facilitates the integration of images as input data. The models we've opted for are indeed representative due to their diverse range of attributes, including visual encoder, LLM backbone, training strategy, image-text data, and instruction-tuning data as illustrated in Table 1 of the main text.  With the rapid development of LVLM, we will involve more models in our LVLM-eHub. Our recent work has included 12 LVLMs including Google's Bard (see [Tiny LVLM-eHub](https://arxiv.org/pdf/2308.03729.pdf) for more details).
>
> **Q2.** [Each model was pretrained by different data, VE, LLM, and adaptor, it may be good to make a roughly fair comparison to provide some findings about the influence of each component.]
>
> **A2.** Thanks for the suggestion.  Since the existing LVLMs are usually different in network structure, training data and training recipes, it would be hard to compare different model components by strict ablation study. However, it is possible to make some rough deductions from our evaluation results.
> First, a strong visual encoder can help extract detailed information from the image, leading to good performance in OCR tasks. For instance, we see that BLIP2, InstructBLIP, and VPGTrans achieve better performance than the remaining 5 LVLMs. This may be because the visual encoder ViT-g/14 used in BLIP2, InstructBLIP, and VPGTrans is more powerful than ViT-L/14 employed in the remaining LVLMs.  Second, prefix-tuning with trainable adaptors (mPLUG-owl and LLaMA-Adapter V2) has demonstrated superior performance in adapting LLMs to multi-modal inputs compared to cross-attention (Otter). From Fig.1 (a) and the updated result of Multimodality Arena in General Response, we see that Otter is inferior to those models with prefix-tuning. It demonstrates that prefix-tuning with trainable adaptors is more appropriate for multi-modality fusion than cross-attention. Third, LLM tuned by instruction following data presents more promising results on Multi-Modality Arena than the LVLM without instruction tuning. From, the updated result of Multimodality Arena in General Response, BLIP2 is always the worst model in the user-end VQA scenario although it achieves good results in quantitative evaluation. Hence, we can conclude that visual instruction tuning is crucial to generate satisfactory answers for open-ended questions.
> Our work takes the first step to understanding the capability of LVLMs in various multimodal applications. In the future, we will investigate the effect of each component should explore various settings with rigorous ablation studies. Our open-source project [LLaMA2-Accessory](https://github.com/Alpha-VLLM/LLaMA2-Accessory.git) can serve this purpose because it unifies various tuning datasets, efficient training techniques, visual encoders, and LLMs into a framework.

---

> > ### Author Response · Authors · 2023-08-15
> > **Response to Reviewer r23C Part (2/2)**
> >
> > **Q3.** [It would be good to have a conclusion, if this benchmark can provide some inspiration about which architecture, which training mechanism, etc, can lead to better performance.]
> >
> > **A3.** Good Question.   We can conclude some actionable insights from our evaluation results.  First, the quality of visual instruction data matters more than quantity in the open-world VQA. We observe that MiniGPT-4, which is tuned by only 3.5K high-quality visual instruction data performs much better than InstructBLIP tuned on visual instruction data adapted from various existing VQA datasets in our MultiModality Arena.  Second,  a strong visual encoder can help extract detailed information from the image, leading to good performance in OCR tasks. For instance, we see that BLIP2, InstructBLIP, and VPGTrans achieve better performance than the remaining 5 LVLMs. This may be because the visual encoder ViT-g/14 used in BLIP2, InstructBLIP, and VPGTrans is more powerful than ViT-L/14 employed in the remaining LVLMs. Third, multi-turn reasoning helps alleviate the hallucination issue, indicating that the evaluation method with critical thinking can induce the correct prediction from the model. We find that LVLM with multi-turn reasoning can determine whether an object exists in the image more accurately than single-turn reasoning. Hence, multi-turn reasoning is appropriate to assess the full potential of the model. Fourth, LVLMs tuned with high-quality instruction-following data present more promising planning ability than models without being tuned with instruction data as demonstrated in Table 7.  We hope that these conclusions can further encourage more insights into developing advanced LVLMs.
> >
> >
> > **Q4.** [One important question could be, after building the benchmark, could the authors provide any potential ways that can improve the performance further?]
> >
> > **A4.** Good Questions. Our evaluation results can inspire further work on developing LVLM in several ways. First, it is important to enhance the diversity of visual instruction data because it helps improve performance in various multimodal tasks. From Fig.A.1 (a) of the Appendix, we see that InstructBLIP achieves competitive results in all six types of multimodal capabilities because InstructBLIP utilizes the most diverse set, drawn from 13 distinct question-answering datasets. In terms of quantitative performance, InstructBLIP stands out with the best zero-shot performance across tasks. Second, the practitioners are also suggested to improve the quality of visual instruction data for developing a competitive LVLM.  Quantity can guarantee performance in many tasks but high quality is crucial for generating satisfactory answers for open-ended questions. Such an observation has already inspired a recent work named Polite-Flamingo [1].  Third, tuning parameters which are significant for LLM can lead to better performance. One ingredient of the adaption module in LLaMA-Adapter V2 is bias-tuning. It tunes the shift and scale parameters which are generally thought to be indispensable for the expressivity of the model [2]. We hope that these conclusions can further encourage more insights into developing advanced LVLMs.
> >
> > [1] Delong Chen et al.  Visual Instruction Tuning with Polite Flamingo.
> > [2] Jonathan Frankle et al. Training BatchNorm and Only BatchNorm. ICLR 2021.

---

### Official Review · Reviewer_5Q2T · 2023-07-21
**Interesting attempt**

**Rating:** 6
**Confidence:** 4
**Clarity:** Yes.

**Strengths:**

Strengths of the article includes:

- Human evaluation
- Interesting capabilities selected for evaluation
- Variety of datasets
- Good selection of existing models for evaluation

**Additional Feedback:**

No additional feedback.

**Correctness:**

Claims are somehow correct because some of the datasets have been already used to train the models.

**Documentation:**

Yes.

**Ethics:**

No.

**Limitations:**

Object hallucination in specific cases is detectable or present in the dataset. However, in some cases, if there is no annotation, it is not clear that how authors claim that they can measure hallucination of a model.

Also it is not clear, that these models are trained on the same datasets that are being used as part of benchmark or not. Authors should provide better evidence that the models did not use those data as part of their training process in any form. For many of the datasets such as COCO it is very likely that the models already saw the data during the training process. This makes the entire benchmark very questionable.

**Opportunities For Improvement:**

Another capability that could have been measured would be image generation capability. Most of the models do not have this capability and they are good at understanding visual data. However, approaches like VisualChatGPT can generate images and also change them with regards to conversational instructions.

**Relation To Prior Work:**

Yes.

**Summary And Contributions:**

Authors provided an evaluation benchmark for large vision language models. Their evaluation includes 16 tasks over more than 40 text related visual datasets to evaluate 6 capabilities of these models. They have built an online with human in loop and offline without human in loop for evaluation.

---

> ### Author Response · Authors · 2023-08-15
> **Response to Reviewer 5Q2T**
>
> Thank you for your valuable comments and insightful suggestions. We have carefully considered each of your queries and addressed them as follows:
>
> **Q1.** [Another capability that could have been measured would be image generation capability. Most of the models do not have this capability and they are good at understanding visual data. However, approaches like VisualChatGPT can generate images and also change them with regard to conversational instructions.]
>
> **A1.** Thank you for your valuable suggestion. We acknowledge the potential for assessing the image generation capability of LVLM models, an aspect which is not extensively explored in recent works. This capability pertains to their aptitude for performing specific tasks through user prompts or even with visual content. The reviewer's recommendation has prompted us to recognize the importance of enhancing this aspect in our future endeavours.
>
> **Q2.** [Object hallucination in specific cases is detectable or present in the dataset. However, in some cases, if there is no annotation, it is not clear that how authors claim that they can measure hallucination of a model.]
>
> **A2.** Thank you for pointing it out. As hallucination problems often refer to cases where the LVLM models tend to generate text with objects that are inconsistent with the given image, it's true that we can not measure the hallucination of a model without annotations. Therefore, properly annotated data is necessary for our evaluation of model hallucination. For our evaluation of object hallucination, we use images in COCO where annotations of objects are included.
>
> **Q3.** [Also it is not clear, that these models are trained on the same datasets that are being used as part of benchmark or not. Authors should provide better evidence that the models did not use those data as part of their training process in any form. For many of the datasets such as COCO it is very likely that the models already saw the data during the training process. This makes the entire benchmark very questionable.]
>
> **A3.** I apologize for any confusion caused. It is true that LVLM models have used images from datasets like COCO. However, the text data used in our evaluation only involves samples from the test set which are not used in the pre-training of LVLMs. Therefore, datasets with COCO's images are still fair to use in our evaluation. However, we note that two VQA datasets used in our evaluation are also part of the visual instruction tuning datasets of InstructBLIP. InstructBLIP's visual instruction tuning datasets include COCO Caption, Web CapFlit, TextCaps, VQAv2, OKVQA, A-OKVQA, OCR-VQA, and LLaVA-Instruct-150K. Indeed, both OK-VQA and OCR-VQA are the held-in split to train models for instruction following, which may lead to the unfairness of evaluation. After excluding the results of OKVQA and OCR-VQA, the average score shown in Table 4 will be modified as Table A below.
>
> **Table A.** Modified Table 4 of the main text after excluding the results of OKVQA and OCR-VQA.
> | Model Name             | BLIP2 | InstructBLIP | LA-V2 | LLaVA | MiniGPT-4 | mPLUG-Owl | Otter | VPGTrans |
> | ---------------------- | ----- | ------------ | ----- | ----- | --------- | --------- | ----- | -------- |
> | Average Score          | 0.771 | **0.879**        | 0.846 | 0.794 | 0.500     | 0.345     | 0.514 | 0.451    |
> | Previous Average Score | 0.758 | **0.900**        | 0.835 | 0.768 | 0.481     | 0.324     | 0.523 | 0.462    |

---

### Official Review · Reviewer_ayea · 2023-07-21
**Paper Review**

**Rating:** 8
**Confidence:** 4
**Correctness:** Yes, I think so
**Clarity:** The areas for improvement in paper cl…

**Strengths:**

- Both benchmarks introduced by the paper constitute important and valuable benchmarks for the academic community.

- The experiments are fairly thorough and well designed -- the use of different evaluation types (question answering, prefix scoring, multi-turn reasoning) for different tasks is a very good design choice.

- The LVLM arena for real-world evaluation is also well designed, and gives a real glimpse into real world utility of these models.

- The authors extensively evaluate eight state-of-the-art LVLMs on both benchmarks.

**Additional Feedback:**

--

**Documentation:**

--

**Ethics:**

No discussion from the authors, but I think that's not warranted for this work.

**Limitations:**

There are no significant limitations. Improvements to the paper writing (and some additional details) are the main areas for improvement.

One limitation (that the authors noted) is that the LVLM arena currently only supports single-turn chats.

**Opportunities For Improvement:**

- The flow of the paper makes things a little difficult to follow, since the reader has to constantly switch contexts between the quantitative benchmark and the online arena. In Section 2.1, for each type of quantitative capability evaluation, the description of which datasets are included should be accompanied by the format of evaluation. This is only described much later, in the zero-shot evaluation subsection and results section (e.g. Section 3.6), which makes the quantitative setup hard to follow. Ideally, the datasets description and evaluation setup (and maybe even results) for the quantitative eval should be grouped together, and the online arena evaluation (and its results) should be a separate section. This would require less context switching between the two evaluation types throughout the paper.

- S2.2: In the online arena evaluation, what kinds of chats/prompts are the users prompted to provide to both models? Is 634 single-round chats a large enough number to draw conclusions? I would hope to see more data points (and hopefully multi-round chats) during the rebuttal period.

- The zero-shot evaluation is a slight misnomer, since some instruction-tuned models (e.g. InstructBLIP) have been trained on many of the tasks. A more appropriate term may be "Direct evaluation".

- It is not clear which tasks utilize the prefix scoring and multi-turn reasoning evaluation type, it would be worth clarifying this.

**Relation To Prior Work:**

Yes

**Summary And Contributions:**

This paper introduces two benchmarks for evaluating the capabilities of large vision-language models (LVLMs). The first is a quantitative capability evaluation that evaluates LVLMs on a set of tasks that capture six different aspects of multimodal capabilities: perception, knowledge acquisition, reasoning, commonsense, embodied intelligence and object hallucination. The second major contribution is an online evaluation arena where users can have conversations with two different models concurrently, and then select their preferred model based on the conversation, thus simulating more realistic scenarios than existing academic benchmarks.

---

> ### Author Response · Authors · 2023-08-15
> **Response to Reviewer ayea**
>
> Thank you for your valuable comments and insightful suggestions. We have carefully considered each of your queries and addressed them as follows.
>
> **Q1.** [S2.2: In the online arena evaluation, what kinds of chats/prompts are the users prompted to provide to both models? Is 634 single-round chats a large enough number to draw conclusions? I would hope to see more data points (and hopefully multi-round chats) during the rebuttal period.]
>
> **A1.** Thank you for your inquiry. In Multi-Modality Arena, users utilize various prompts to get answers from both models. For example, they would upload an image of a landmark in a city and ask "Do you know which city it is?". Regarding the quantity of data points employed to compile the leaderboard, the updated table is presented in Table A. Within this table, the designations "paper" and "web" pertain to the outcomes showcased in our paper and on our website, encompassing 634 and 1770 data points, respectively. The most recent results are denoted as "latest," incorporating a total of 2750 data points. Moreover, we are pleased to announce the availability of multi-round chats on our [Multi-Modality Arena page](http://vlarena.opengvlab.com/). We warmly welcome any feedback or commentary on this new feature.
>
> **Table A.** The updated results of Multi-Modality Arena. We see that model's ranking can converge with minor changes when more samples are collected.
> | Paper (634)  |            | Web (1770)   |            | Latest (2750) |            |
> | ------------ | ---------- | ------------ | ---------- | ------------- | ---------- |
> | Model Name   | Elo rating | Model Name   | Elo rating | Model Name    | Elo rating |
> | mPLUG-Owl    | 1027.0     | LA-V2        | 1023       | LA-V2         | 1038.74    |
> | MiniGPT-4    | 1021.3     | LLaVA        | 1019.9     | MiniGPT-4     | 1009.87    |
> | Otter        | 1013.2     | VPGTrans     | 1012.1     | LLaVA         | 1002.76    |
> | LA-V2        | 1010.2     | MiniGPT-4    | 1011.9     | VPGTrans      | 999.63     |
> | LLaVA        | 1009.7     | InstructBLIP | 999.5      | InstructBLIP  | 995.45     |
> | InstructBLIP | 1003.7     | mPLUG-Owl    | 996.3      | mPLUG-Owl     | 985.60     |
> | VPGTrans     | 974.3      | Otter        | 981.5      | Otter         | 972.11     |
> | BLIP2        | 949.4      | BLIP2        | 955.8      | BLIP2         | 948.8      |
>
> **Q2.** [The zero-shot evaluation is a slight misnomer, since some instruction-tuned models (e.g. InstructBLIP) have been trained on many of the tasks. A more appropriate term may be "Direct evaluation".]
>
> **A2.** Thank you for your suggestion. We agree that it is more rigorous to use the term "direct evaluation" than the term "zero-shot" evaluation. However, the context of zero-shot generalization did change when it comes to the era of LLMs. Originally, a model trained on dataset A is evaluated on dataset B that is different from dataset A via zero-shot prompting, i.e., direct evaluation without fine-tuning on or having seen any data point from dataset B. However, nowadays, foundation models are generally pre-trained on a large corpus of datasets which may be huge enough to contain some downstream dataset samples somehow. Therefore, as stated in GPT-4 technical report, it performs explicit filtering to avoid data contamination via simple string matching before evaluation on downstream datasets of interest, while simple string matching still cannot handle cases where the original text is changed in form, such as paraphrasing in general. Nevertheless, OpenAI and many other researchers still use the term zero-shot evaluation in general, while having implicit consent about issues mentioned above.
>
> **Q3.** [It is not clear which tasks utilize the prefix scoring and multi-turn reasoning evaluation type. It would be worth clarifying this.]
>
> **A3.** Thank you for your inquiry. In Table 5, we employ the "Prefix Scoring" method for the "Visual Commonsense" task on the ImageNetVC dataset. For the same "Visual Commonsense" task, we utilize the "Multi-turn Reasoning" approach on the VCR dataset, as presented in Table 5. Additionally, this "Multi-turn Reasoning" technique is applied to the "Visual Reasoning" task using the SNLI-VE dataset, as outlined in Table 4.

---

### Official Review · Reviewer_j6vp · 2023-07-26
**Large benchmark suite for text-generative vision-language models**

**Rating:** 8
**Confidence:** 4
**Correctness:** Yes

**Strengths:**

- The LVLM Arena is an innovative way to judge LVLMs, assuming enough users participate.
- The evaluation is extremely broad with 47 benchmarks.
- The 8 analysed models are well compared in their architecture and setup.
- The final analysis and comparison of the models is very informative.

**Additional Feedback:**

Line 267 typo conclusion

**Clarity:**

Remaining questions:

- Why is VQAv2 not used, the most popular VQA benchmark? It is not mentioned in the paper at all (apologies in case I missed it).

- Line 154: How is multi choice evaluated exactly? Getting the likelihood of an image-text pair is easy with CLIP, however it is non-trivial with a decoder models like Llava or InstructBLIP, since the LLM will return an individual logit for each output token. How does obtaining this image-text pair likelihood look like mathematically?

- Details on the arena: How are the images and questions selected or created? Will all models eventually see the same images and questions, thus making the comparison fair?

- Line 157: How does the multi-turn reasoning look like exactly? Which models are used for which turns? The text is rather vague on this.

**Documentation:**

Yes

**Limitations:**

- 634 annotations in the LVLM Arena for 8 models is few datapoints. It is not guaranteed that these findings are representative for model strengths.

- There is few detailed analysis on the tasks. The benchmark suite is very broad but not in depth for the individual tasks. However, due to having 47 tasks it is not possible to spend months of time on each task. This limitation is not mentioned in the paper.

**Opportunities For Improvement:**

- The split into the various parts like visual perception, visual knowledge acquisition etc. in chapter 2.1 seems somewhat arbitrary. For example: Counting "2 dogs" is perception, creating the caption "A photo of 2 dogs" is knowledge acquisition. Maybe more finegrained groups like classification, counting, OCR instead of the aggregated groups could be better.

**Relation To Prior Work:**

Yes

**Summary And Contributions:**

Authors propose a new benchmark suite to evaluate large vision-language models. They aggregate many existing benchmarks and group them into 6 groups. Also, they propose an online user-based arena. They evaluate 8 of the most popular LVLMs on their benchmark suite.

---

> ### Author Response · Authors · 2023-08-15
> **Response to Reviewer j6vp Part (1/2)**
>
> Thank you for your valuable comments and insightful suggestions. We have carefully considered each of your queries and addressed them as follows.
>
> **Q1.** [The split into the various parts like visual perception, visual knowledge acquisition etc. in chapter 2.1 seems somewhat arbitrary. For example: Counting "2 dogs" is perception, and creating the caption "A photo of 2 dogs" is knowledge acquisition. Maybe more fine-grained groups like classification, counting, and OCR instead of the aggregated groups could be better.]
>
> **A1.** Thank you for asking your question. It's true that using more detailed groups can provide a clearer picture of the performance of LVLM models compared to the aggregated group we previously used. Table A shows the updated model performance scores with the more detailed groups.
>
> **Table A.** The performance of LVLMs  within the more detailed groups in  visual perception and visual knowledge acquisition.
> | Task      | BLIP2 | InstructBLIP| LA-V2| LLaVA| MiniGPT-4|mPLUG-Owl|Otter|VPGTrans|
> | ----------- | ----------- |----------- |----------- |----------- |----------- |----------- |----------- |----------- |
> ImgCls | 0.893|**0.976**|0.881|0.618|0.779|0.814|0.538|0.475|
> OC|0.805|**0.834**|0.716|0.510|0.521|0.391|0.972|0.480|
> MCI|0.925|**1.000**|0.803|0.720|0.846|0.326|0.628|0.598|
> OCR|0.958|**1.000**|0.355|0.319|0.268|0.261|0.193|0.605|
> KIE|0.681|**0.738**|0.611|0.502|0.333|0.150|0.498|0.627|
> Captioning|0.960|**0.980**|0.582|0.038|0.086|0.003|0.189|0.603|
>
> **Q2.** [634 annotations in the LVLM Arena for 8 models is few datapoints. It is not guaranteed that these findings are representative for model strengths.]
>
> **A2.** Thank you for your inquiry. With the successful onboarding of our online arena, we've been able to accumulate an impressive dataset comprising more than 634 data points. The updated LVLM Arena Leaderboard is now presented in Table B. Please take note that under the "paper" category, we've included the results featured in our paper based on 634 data points. Additionally, the "web" category encompasses results displayed on [our website](http://lvlm-ehub.opengvlab.com/blog_leaderboard.html), drawn from a larger dataset of 1770 data points. Lastly, the "latest" category showcases the most current leaderboard, incorporating all 2750 data points we've gathered thus far.  We see that model's ranking can converge with minor changes when more samples are collected.  In particular, LA-V2,  LLaVA,  MiniGPT-4, and VPGTrans are more competitive than InstructBLIP, mPLUG-Owl, Otter, and BLIP2. We see that model's ranking can converge with minor changes when more samples are collected.
>
> **Table B.** The results of Multi-Modality Arena in different timeframes.
> | Paper (634)  |            | Web (1770)   |            | Latest (2750) |            |
> | ------------ | ---------- | ------------ | ---------- | ------------- | ---------- |
> | Model Name   | Elo rating | Model Name   | Elo rating | Model Name    | Elo rating |
> | mPLUG-Owl    | 1027.0     | LA-V2        | 1023       | LA-V2         | 1038.74    |
> | MiniGPT-4    | 1021.3     | LLaVA        | 1019.9     | MiniGPT-4     | 1009.87    |
> | Otter        | 1013.2     | VPGTrans     | 1012.1     | LLaVA         | 1002.76    |
> | LA-V2        | 1010.2     | MiniGPT-4    | 1011.9     | VPGTrans      | 999.63     |
> | LLaVA        | 1009.7     | InstructBLIP | 999.5      | InstructBLIP  | 995.45     |
> | InstructBLIP | 1003.7     | mPLUG-Owl    | 996.3      | mPLUG-Owl     | 985.60     |
> | VPGTrans     | 974.3      | Otter        | 981.5      | Otter         | 972.11     |
> | BLIP2        | 949.4      | BLIP2        | 955.8      | BLIP2         | 948.8      |
>
> **Q3.** [Why is VQAv2 not used, the most popular VQA benchmark? It is not mentioned in the paper at all (apologies in case I missed it).]
>
> **A3.** Good question. VQAv2 is definitely one of the most popular VQA benchmarks. However, InstructBLIP explicitly states that it uses the VQAv2 dataset in the held-in split to train the model. Therefore, in order to avoid data contamination and ensure fairness among models, we exclude VQAv2 in our evaluation benchmark suite.

---

> > ### Author Response · Authors · 2023-08-15
> > **Response to Reviewer j6vp Part (2/2)**
> >
> > **Q4.** [Line 154: How is multi choice evaluated exactly? Getting the likelihood of an image-text pair is easy with CLIP, however it is non-trivial with a decoder models like Llava or InstructBLIP, since the LLM will return an individual logit for each output token. How does obtaining this image-text pair likelihood look like mathematically?]
> >
> > **A4.** Thank you for your insightful question. In our experiments, most VQA-like multi-choice datasets are evaluated by employing the "word matching" method, i.e., whether the ground-truth answer exists in models' generations. For Line 154 "prefix-based score" specifically, it feeds the same image together with each option of a multi-choice visual question respectively and gathers the logarithm of probabilities of each token of the candidate option by summing and then optionally normalizing over the number of tokens of that option, which finally results in a holistic loglikelihood of that option given the image. For more details, please refer to [EleutherAI/lm-evaluation-harness](https://github.com/EleutherAI/lm-evaluation-harness/tree/e47e01beea79cfe87421e2dac49e64d499c240b4) and https://huggingface.co/blog/evaluating-mmlu-leaderboard#now-how-do-we-evaluate-the-model-from-these-prompts. We also present the formulation of the prefix-based score for multi-choice VQA in Sec.A.3 of the supplementary revision.
> >
> > ```Python
> > # PyTorch-like pseudo code
> > prompt = "{prompt}"
> > prompt_token_ids = tokenizer(prompt)
> > option = "{option}"
> > option_token_ids = tokenizer(option)
> > token_ids = prompt_token_ids + option_token_ids
> >
> > inputs = token_ids[:-1]
> > targets = token_ids[1:]
> > logits = llm(inputs) # [len(inputs), vocab_size]
> >
> > # only keep continuation for the prompt
> > logits = logits[-len(option_token_ids):]
> > logprobs = F.log_softmax(logits)
> > logprobs_option = torch.gather(logprobs, 1, option_token_ids)
> >
> > logprobs_option_sum = logprobs_option.sum()
> > # optionally perform normalization
> > logprobs_option_sum_normalized = logprobs_option_sum / len(option_token_ids)

---

> > > ### Comment · Reviewer_j6vp · 2023-08-25
> > >
> > > Thank you alot for your insightful response. I have decided to keep my rating at 8 and increase my confidence to 4. I hope your paper will be accepted.

---

### Author Response · Authors · 2023-08-15
**General Repsonse**

We are thankful to the reviewers for their detailed reviews and thoughtful suggestions on our manuscript. Notably, we are glad that the reviewers agreed that the LVLM Arena is an innovative way to judge LVLMs, that the evaluation is extremely comprehensive, that interesting capabilities were selected for evaluation, and that the findings are informative.

In general, the reviewers also raise some concerns about our work, including 1) more details about LVLM Arena; 2) more details about the evaluation process, and 3) how our evaluation results inspire future work on developing LVLMs, and so on. During the rebuttal, we address these concerns through additional experiments and clarification. We are happy to run more experiments if the reviewer has any piece of interest.

---

### Decision · Program_Chairs · 2023-09-22

**Decision:**

Reject

**Comment:**

This paper presents a novel and interesting evaluation framework for large visual-language models (LVLM). Reviewers' judgements have been split, but after the rebuttal, most concerns have been addressed by the authors. For instance, two reviewers concerned 634 single-round chats are not large enough, and the authors have provided larger data points (1770 to 2750). Compositionality was asked, and the authors provided additional benchmarks. Two reviewers also asked some insights or conclusion about the benchmark results, which are well-answered in the rebuttal.

I still have two concerns:
 - A reviewer asked representativeness of the 8 models selected, and the authors answered they chose "most up-to-date, open-source, and re-implementable" models. This does not fully address the question, in my opinion, since these small number of models then may not be representative of LVLMs. The authors may still need to provide a full list of LVLM models to be considered and detailed reasons for why each model has been ignored in this study.

 - More importantly, the authors conducted user study without IRB approval. The reviewer Z9KL pointed this out and asked explicitly if they have been approved by IRB, and the authors simply answered "We do not necessarily have IRB Approval." Considering the significance of user participation of this arena and user study, however, we do believe that it is critical to have IRB approval for this study. For this reason, we recommend rejection of this paper, although we do see the value of this work, as most reviewers marked. We recommend the authors to resubmit after having the required ethical approvals including IRB.